# Beyond Audio-Visual Alignment: Unmasking Talking Head Deepfakes via Red Hue Discrepancies in HSV Color Space

## Abstract

Deepfake video detection is crucial in preventing the dissemination of harmful forged audio-visual content. However, the lack of radiance field-based videos in current audio-visual forgery datasets presents a limitation that impedes comprehensive evaluation of detection models. To address this issue, we introduce **R**adiance **F**ield **A**udio-**V**isual (RFAV) dataset, comprising fake videos synthesized using Neural Radiance Fields (NeRF) and 3D Gaussian Splatting (3DGS), to fill this data gap. As for detection models, existing methods primarily focus on audio-visual mismatches and demonstrate limited effectiveness when applied to forged videos with highly synchronized lip movements. To address this challenge, we re-think talking head deepfakes from a novel perspective based on the distribution of red hue in the HSV color space. We find real and forged videos exhibit distinct differences in the HSV color space, particularly in regions of intense facial motion. Based on this observation, we propose a **R**ed **H**ue-based **T**alking **H**ead **F**orgery **D**etection (RHTHFD) model. This unsupervised learning framework employs visual region attention to adaptively fuse HSV and visual features, while integrating re-weighted speech features to improve the generalization of deepfake detection. Our method achieves state-of-the-art performance on multiple evaluation benchmarks, including the proposed radiance field-based RFAV dataset.

## 1 Introduction

With the rapid advancement of Generative Adversarial Networks (GANs) Luo et al. (2025), Diffusion Models (DMs) Dewan et al. (2024), and Radiance Fields (RFs) Kerbl et al. (2023); Xie et al. (2025), audio-driven talking head generation Liu et al. (2025b) has become increasingly sophisticated, yielding greater realism. High-fidelity audio-visual synthesis brings notable benefits to society Ma et al. (2023), but also introduces serious risks such as identity theft, conspiracy theories, and political misinformation Zhang et al. (2025). Therefore, it is invaluable to develop effective methods for distinguishing AI-generated fake videos from authentic real-world footage.

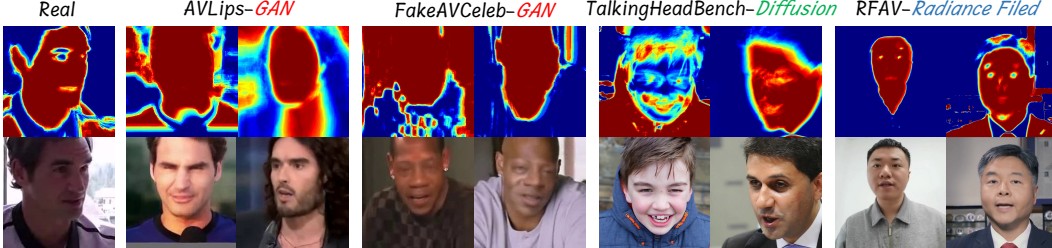

Figure 1: Visualization of the red hue in HSV color space for audio-visual real and fake videos. Forgery methods show noticeable color bias and detail loss in the red hue, resulting in a marked discrepancy from the authentic distribution.

Early deepfake detectors Cai et al. (2023b); Zheng et al. (2021) focus on visual features to identify facial manipulations, but they fall short in detecting talking head forgeries. With the emergence

of audio-visual datasets, integrating speech and visual features has become a mainstream method Feng et al. (2023); Oorloff et al. (2024) for forgery detection. Recent studies focus on learning audio-visual alignment representations Liu et al. (2024); Smeu et al. (2025) and have achieved some improvements. However, the generalization of these methods remains challenged by highly synchronized forged videos and cross-dataset evaluation scenarios.

Unlike deepfakes that replace the entire face or manipulate facial attributes Korshunova et al. (2017), talking head deepfakes primarily generate facial dynamics synchronized with speech, making detection particularly challenging due to their localized modifications. In response to this threat, an increasing number of facial deepfake benchmarks Khalid et al. (2021); Hou et al. (2024) have been proposed. Most existing benchmarks rely on GANs or pre-trained lip-sync generators Liu et al. (2024), primarily focusing on easily detectable facial forgery techniques Rossler et al. (2019). A recent benchmark Xiong et al. (2025) built on cutting-edge diffusion models improves facial realism and enriches data diversity. However, as a critical component of talking head generation, RFs Liu et al. (2025a); Bao et al. (2025) are not included in existing benchmarks, thereby limiting the comprehensiveness of current detection evaluations. To address this limitation, we propose a novel

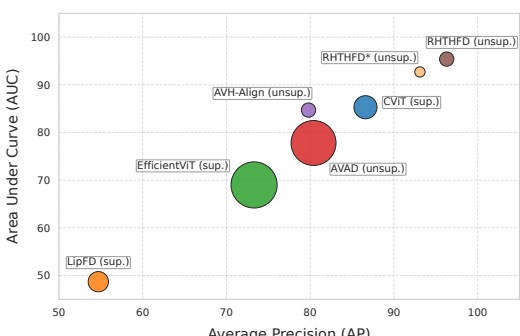

Figure 2: Our RHTHFD achieves the highest AUC (%) while maintaining a superior AP (%). Notably, RHTHFD*, which is trained on only half the data used by RHTHFD, still significantly outperforms all baseline methods. The size of each bubble represents the number of training videos used by each model. The larger the bubble, the more data it indicates.

**R**adiance **F**ield **A**udio-**V**isual (RFAV) benchmark developed using neural radiance fields-based SyncTalk Peng et al. (2024) and 3d gaussian splatting-based GaussianTalker Cho et al. (2024). These two methods ensure that the benchmark remains timely and challenging. As shown in Table 3, we compare the AUC and AP of existing deepfake detection models Coccomini et al. (2022); Liu et al. (2024); Smeu et al. (2025); Feng et al. (2023); Wodajo & Atnafu (2021) with our method on the proposed RFAV dataset. The results demonstrate that both supervised and unsupervised models exhibit performance degradation, with a detector performing even worse than random guessing.

Most existing forgery detectors rely on audio-visual alignment features, proving inadequate when facing highly realistic synthetic videos. To address this issue, we reconsider talking head forgery detection from the perspective of the red hue in the HSV color space. In Figure 1, fake videos generated by GANs, DMs, and RFs tend to exhibit either excessive detail or noticeable loss in the mouth region within the HSV color space, making them more distinguishable from real videos. In addition, other facial regions such as the eyes and nose also exhibit differences. These anomalies underscore the limitations of current forgery methods in reconstructing fine-grained visual features, particularly in color-sensitive regions, offering a basis for our forgery detection strategy. The results in Figure 2 show that incorporating HSV features enables our method to significantly outperform all other detectors, even with minimal training data. Although HSV is widely used in face analysis, its application to talking head deepfake detection has not yet been explored. We leverage salient irregularities in the red hue to reveal forgery artifacts across different synthesis paradigms, thereby improving detection performance.

In this paper, we propose an unsupervised learning framework, **R**ed **H**ue-based **T**alking **H**ead **F**orgery **D**etection (RHTHFD), which leverages the red hue of HSV features to realize generalized talking head forgery detection. Specifically, we first map the entire video sequence into the HSV color space, define the red hue, and apply red masks to extract temporally coherent color features. A pre-trained DinoV2 model Oquab et al. (2023) is employed to extract global and local features from the averaged HSV features. Then, a visual region attention and an adaptive weight algorithm are designed to fuse HSV-based local-global features with visual representations extracted by the lip-reading model Shi et al. (2022). Finally, an adaptive weighting module is introduced to fuse the enhanced visual features with audio features that exhibit visual consistency, thereby generating misalignment scores for effective forgery detection. Experimental results demonstrate that the proposed

method achieves superior performance on audio-visual benchmarks constructed using GANs, DMs, and the newly introduced RFs, consistently surpassing existing methods, while also maintaining stable performance under different perturbation attacks.

Our contributions can be summarized as follows:

- We introduce a novel perspective for talking head forgery detection by analyzing the differences between real and forged videos in the red hue of HSV color space. A visual region attention and an adaptive weight generator are also designed to effectively fuse HSV features into the audio-visual stream.
- We introduce the RFAV dataset, a carefully curated benchmark designed to address the lack of radiance field-generated talking heads in existing benchmarks.
- Extensive experimental results demonstrate that our model exhibits strong generalization against forged videos generated by GANs, DMs, and RFs, significantly outperforming baseline methods.

## 2 RELATED WORK

### 2.1 TALKING HEAD DEEPFAKE DETECTION

Talking head generation aims to map acoustic features to temporally aligned facial motions, while learning interactive information from multimodal spaces Chen et al. (2025); Wei et al. (2023). With the growing challenges posed by the rapid development of this technology, researchers are shifting their focus from unimodal feature learning Huang et al. (2023); Zheng et al. (2021) to audio-visual collaborative detection Liu et al. (2024); Yang et al. (2023). Several methods directly train audio-visual networks using labels from real and forged videos Chugh et al. (2020); Mittal et al. (2020), while others employ audio-visual self-supervised learning to pre-train models Zeng et al. (2021); Haliassos et al. (2022); Zhou & Lim (2021), followed by fine-tuning with labeled data. However, all these methods operate within a supervised learning framework, limiting the robustness of the detection models. To enhance the generalization capability, an alternative method involves unsupervised learning strategies that rely solely on real data for training Li et al. (2024b); Ricker et al. (2024). AVAD Feng et al. (2023) proposes an anomaly-based video detection method that trains an autoregressive model on features capturing temporal synchronization between video and audio. Similarly, AVH-Align Smeu et al. (2025) improves the robustness of deepfake detection by leveraging self-supervised audio-visual representations and reducing reliance on dataset-specific biases. Although these methods enhance detection performance by leveraging audio-visual alignment, they remain ineffective against talking head generation models that exhibit highly consistent lip synchronization. In contrast, our method analyzes the distribution of the red hue in HSV color space to capture the distinctive characteristics of fake videos, while also considering audio-visual consistency for talking head deepfake detection.

### 2.2 AUDIO-VISUAL DEEPFAKE DATASETS

Early deepfake datasets primarily contain synthetic videos based on face-swapping methods Rosberg et al. (2023), resulting in easily detectable artifacts such as boundary inconsistencies. Talking head generation aims to manipulate lip movements and facial expressions while maintaining the overall identity of the person, increasing the difficulty of deepfake detection. Most datasets such as FakeAVCeleb Khalid et al. (2021) and FaceForensics++ Rossler et al. (2019) collect fake videos generated by GAN-based models Prajwal et al. (2020) or derived from limited facial manipulation methods like Face2Face Thies et al. (2016).Recent datasets including AVLips Liu et al. (2024), LAV-DF Cai et al. (2023a), AV-DeepFake1M Cai et al. (2024), and PolyGlobFake Hou et al. (2024)

Table 1: Summary of selected identities in the RFAV dataset. The five identities are carefully chosen to cover diversity in age, gender, skin tone, and language.

| ID | Gender | Skin | Language | Age |
|------|--------|--------|----------|-------|
| ID1 | Male | Yellow | Chinese | 25–30 |
| ID2 | Male | White | French | 45–50 |
| ID3 | Male | Black | English | 50–55 |
| ID4 | Male | Yellow | English | 55–60 |
| ID5 | Female | White | English | 45–50 |

employ talking head generation with robust lip-sync performance, posing significant challenges to existing detection frameworks. However, the lack of rigorous data curation in these datasets often results in the inclusion of numerous low-quality samples. More recently, TalkingHeadBench Xiong et al. (2025) incorporates diffusion-based generation methods into the construction of deepfake datasets, enhancing both the quality and diversity of synthetic heads. Although radiance fields play a pivotal role in talking head generation, they are overlooked in existing deepfake datasets. Undoubtedly, the absence of this synthesis paradigm hinders the comprehensiveness of detection method evaluation. To fill this gap, we select high-quality radiance field synthesis methods based on NeRF Peng et al. (2024) and 3DGS Cho et al. (2024) to construct a dataset RFAV. Experimental results indicate that existing methods struggle to achieve satisfactory performance on our dataset.

## 3 RFAV DATASET

Most audio-visual datasets are constructed using GANs or diffusion-based methods, while data synthesized via radiance fields remains largely absent. To address this gap, we introduce a high-quality Radiance Field Audio-Visual (RFAV) dataset. As shown in Figure 3, it comprises 6,000 forged videos generated using state-of-the-art NeRF and 3DGS methods Peng et al. (2024); Cho et al. (2024). The primary objective of RFAV is to evaluate the generalization of existing audio-visual detectors, thereby enabling a more sane assessment of progress in deepfake detection.

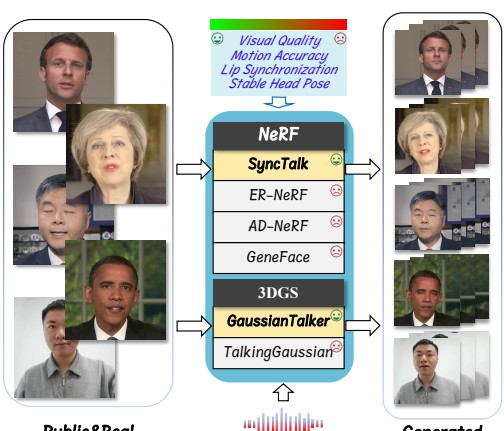

Figure 3: Utilizing NeRF and 3DGS methods that demonstrate superior performance in visual quality, motion accuracy, lip synchronization, and head dynamics, RFAV comprises high-fidelity radiance field-based talking heads.

To ensure high-quality data, we compare the synthesis performance of recent radiance field methods in terms of visual quality, motion accuracy, and audio-visual consistency. In addition, we manually filter out methods that exhibit issues such as jitter, which are difficult to capture through quantitative evaluations. After a comprehensive comparison of six reproducible methods (AD-NeRF Guo et al. (2021), ER-NeRF Li et al. (2023), GeneFace Ye et al. (2023), SyncTalk Peng et al. (2024), TalkingGaussian Li et al. (2024a) and GaussianTalker Cho et al. (2024)), we ultimately select NeRF-based SyncTalk and 3DGS-based GaussianTalker. Both methods perform well across multiple evaluation criteria. Radiance field methods typically require training speaker-specific models, which makes data collection extremely challenging. Therefore, we perform customized training on both commonly used portraits in radiance fields and real-world individuals. These pre-trained models are then driven by real and fake audios from Speech-forensics Ji et al. (2024) to synthesize talking head deepfake videos. As shown in Table 1, RFAV includes five carefully selected identities covering different ages, genders, skin tones, and languages, each driven by a large number of real and synthetic speech samples. This design ensures the dataset still provides rich variations in facial dynamics. This dataset is constructed to fill the gap in radiance-field–based talking head data, thereby enabling a more comprehensive evaluation of detector performance.

## 4 METHOD

We propose an HSV red hue-based unsupervised learning framework, RHTHFD, as shown in Figure 4. First, HSV features are extracted from the video frames, which are then processed by DinoV2 Oquab et al. (2023) to obtain global and local semantic representations. With the proposed visual region attention, these features are fused with vanilla features extracted by the visual encoder of a lip-reading network Shi et al. (2022). Finally, after combining the speech features, an MLP predictor is used to perform fake detection.

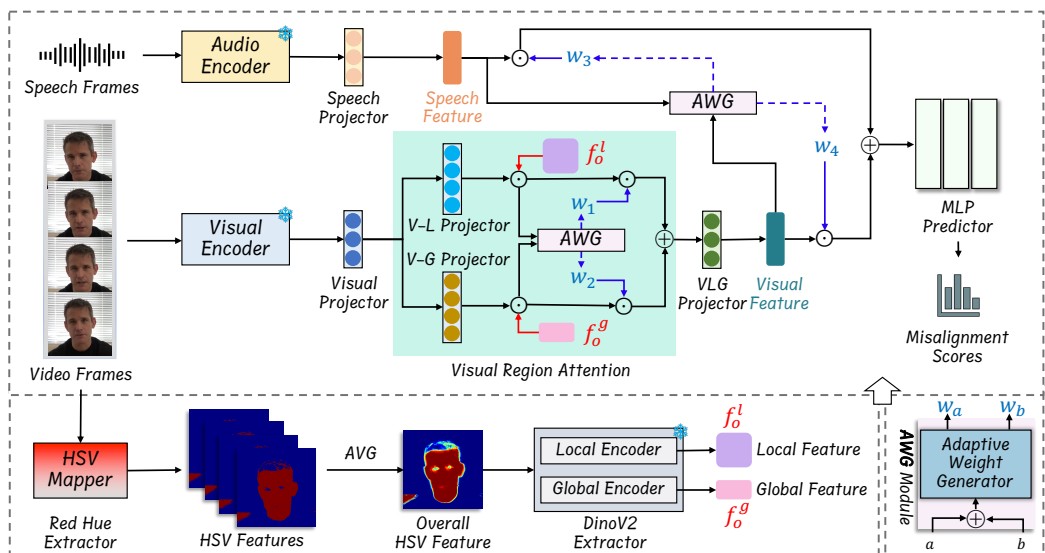

Figure 4: Overview of our proposed RHTHFD. We use the visual and audio encoders of a pre-trained lip-reading network to extract vanilla visual features and temporally aligned speech features, respectively. To enable the model to effectively learn how to distinguish fake videos from the perspective of HSV color distribution, we employ DinoV2 to extract global and local semantic representations. A visual region attention mechanism is applied to fuse the semantics of the visual stream, while AWG module automatically learns feature weights based on different modalities.

## 4.1 RED HUE IN HSV COLOR SPACE

Visual perception plays a decisive role in distinguishing between real and fake data Jia et al. (2025). Hemoglobin strongly reflects red light, producing stable and structured red hue patterns in real human skin Weyrich et al. (2006). These patterns encode fine-grained texture and blood-flow dynamics that are difficult to replicate. In contrast, generative models often introduce subtle color inconsistencies due to limitations in modeling skin tone and albedo Jain et al. (2022); Perera & Patel (2023); Zeng & Kalantari (2025). As a result, synthetic talking head videos often show deviations in color patterns compared to real videos, which may manifest as unnatural uniformity or blurred facial details. To explore the differences in feature distribution between fake and real videos, we map the red hue in HSV color space Sural et al. (2002). Figure 1 presents HSV-based red hue visualizations of randomly selected fake videos from GAN-based AVLips Liu et al. (2024) and FakeAVCeleb Khalid et al. (2021), diffusion-based TalkingHead-Bench Xiong et al. (2025), and the proposed radiance fields-based RFAV. In the HSV color space, GAN-generated videos often lack distinct facial features, diffusion-based synthesis produces overly refined facial details, and radiance field reconstructions suffer from overly smooth facial details.

We also compute the average red hue distributions across all real and forged videos and measured their differences using KL divergence and Earth Mover's Distance (EMD). In Table 2, we find GAN forgeries are closest to real data but still exhibit local irregularities around key facial regions (Figure 1). Diffusion and radiance field forgeries show larger distributional shifts. Moreover, comparisons across forgery methods reveal distinct red hue signatures for each paradigm. These observed differences can be explained by the inherent characteristics of generative models. GAN-based forgeries usually preserve coarse color statistics but struggle

Table 2: KL divergence and Earth Mover's Distance (EMD) between real and synthetic talking head videos across different generation methods.

| Method | KL | EMD |
|---|---|---|
| Real vs GAN | 0.0797 | 0.9918 |
| Real vs Diffusion | 0.1445 | 3.4803 |
| Real vs Radiance Field | 0.6755 | 5.1119 |
| GAN vs Diffusion | 0.1044 | 3.0368 |
| GAN vs Radiance Field | 0.5598 | 4.6323 |
| Diffusion vs Radiance Field | 0.2667 | 1.8901 |

to reproduce fine local skin textures and microvascular structures, leading to subtle irregularities in the red hue within key facial regions. Diffusion models often introduce more variations in color and

lighting, resulting in larger distributional shifts. Radiance-field–based methods rely on 3D reconstruction and rendering pipelines, and due to imprecise modeling of skin reflectance and lighting, they may amplify artifacts in hue consistency. By comparing differences in the red hue between fake and real videos, we introduce a novel and effective feature for AI-generated video detection models. The overall HSV feature of each video is computed as follows:

$$f_o = \frac{1}{n} \sum_{i=1}^{n} HSV_{red}(V_i), \tag{1}$$

where $n$ is the frame number of the target video $V$. Directly applying HSV feature to model learning may lead to overfitting on superficial color statistics, making it difficult to capture the spatial-semantic relationships within facial structures. To address this limitation, we use DinoV2 to extract global and local semantic representations from $f_o$. The global feature $f_o^g$ model the consistency of facial coloration, while the local feature $f_o^l$ decode fine-grained patterns in deformation-dense regions.

## 4.2 VISUAL REGION ATTENTION

To ensure temporal consistency between visual and speech features, we adopt a pre-trained lip-reading network Shi et al. (2022) to represent both the speech and visual content of the same video. This model is a transformer-based architecture trained with a self-supervised learning strategy, which demonstrates strong performance even under challenging audio-visual conditions Wang et al. (2023). Its modality selector supports audio-only, video-only, and audio-visual modes by masking unused modalities, and we use the first two to extract speech features $f_s$ and vanilla visual features $f_v$ with corresponding speech and visual projectors.

The global and local HSV features extracted by DinoV2 rarely affect the visual stream equally. Hence, learning how these features influence different regions of $f_v$ is crucial for improving the performance of forgery detection. To capture the relationship between discriminative regions (e.g., mouth) in the HSV features and the visual representation $f_v$, we introduce a visual region attention mechanism, as shown in Figure 4. Specifically, we first employ the V-G and V-L projectors to extract global and local content features $f_v^g$ and $f_v^l$ from $f_v$. The global and local HSV features are then treated as regional attention vectors, which are used to reweight each channel of $f_v^g$ and $f_v^l$, respectively. Given that local and global features contribute differently to the visual stream, we apply an Adaptive Weight Generator (AWG) module to the reweighted features to learn region-specific importance weights $w_1$ and $w_2$. AWG consists of four linear layers with ReLU activations, followed by a softmax function to produce normalized weights. Finally, the learned global and local features are concatenated to form a unified visual feature $f_v^{lg}$, which integrates region-aware semantic information. The overall process is as follows:

$$f_v^{lg} = \left(f_o^l \odot f_v^l\right) w_1 \oplus \left(f_o^g \odot f_v^g\right) w_2, \tag{2}$$

where $w_1 + w_2 = 1$, $\odot$ denotes the Hadamard product, and $\oplus$ refers to the concatenation operator. The importance of speech and visual features is inherently imbalanced in forgery detection. Therefore, before fusing the speech feature $f_s$ and the visual feature $f_v^{lg}$ for obtaining misalignment score, we employ the other AWG module to differentiate their relative importance.

## 4.3 TRAINING OBJECTIVE

Inspired by Feng et al. (2023), we maximize the temporal alignment probability between audio frames and the corresponding visual frames to train our detection network $\Psi$. Similar to the contrastive loss InfoNCE Wang et al. (2023), we define the final loss function as the negative average probability over the entire video:

$$L = -\frac{1}{F} \sum_{i=1}^{F} log \frac{exp\Psi_{ij}}{\sum_{k \in T(i)} exp\Psi_{ik}}, \tag{3}$$

where $T_{(i)}$ denotes the temporal neighborhood of the $i^{th}$ frame, and $F$ is the frame number. Once training is completed, we can estimate a fakeness score for each audio-visual frame pair, where well-aligned audio and visual frames tend to produce lower scores. We apply a smoothed max function to aggregate the frame-level scores and compute the overall alignment score for the entire video.

Table 3: Results on the AVLips, FKAV, RFAV, and THB datasets. We report AP (%) and AUC (%) with the best results in **bold** and the second-best results underlined. *Training Videos* reflects the training data volume. *V* and *A* denote visual and audio modalities, respectively. *sup.* and *unsup.* refer to supervised and unsupervised methods. We use released checkpoints for supervised baselines. Since AVAD has no training code, we evaluate it using its provided weights. AVH-Align is retrained under the same settings for fair comparison. The training data for RHTHFD* is half the size of that used for RHTHFD.

| Methods | Type | Modality | Training Videos | AVLips | | FKAV | | RFAV | | THB | |
|---------|------|----------|------------------|--------|------|------|------|------|------|------|------|
| | | | | AP | AUC | AP | AUC | AP | AUC | AP | AUC |
| CViT | Sup. | V | 27k+ | 63.5 | 63.1 | 91.1 | 88.5 | 86.6 | 85.3 | 44.5 | 42.1 |
| EfficientViT | Sup. | V | 126k+ | 63.3 | 64.8 | 95.1 | 90.9 | 73.3 | 69.0 | 31.6 | 21.7 |
| LipFD | Sup. | AV | 10k+ | 85.3 | 84.7 | 83.4 | 77.0 | 54.7 | 48.7 | 45.0 | 49.2 |
| AVAD | Unsup. | AV | 120k+ | 76.5 | 73.2 | 92.1 | 84.8 | 80.4 | 77.8 | 43.8 | 48.1 |
| AVH-Align | Unsup. | AV | 2k | 74.3 | 84.5 | 93.5 | 93.0 | 79.8 | 84.7 | 64.8 | 82.3 |
| RHTHFD (Ours) | Unsup. | AV | 2k | **95.6** | **95.9** | **95.9** | **95.7** | **96.3** | **95.4** | **89.1** | **93.0** |
| RHTHFD* (Ours) | Unsup. | AV | 1k | 93.0 | 93.6 | 94.7 | 93.7 | 93.1 | 92.7 | 85.9 | 91.8 |

## 5 EXPERIMENTS

### 5.1 EXPERIMENTAL SETUP

**Datasets and Metrics.** We conduct experiments on four datasets: the publicly available AVLips Liu et al. (2024), FakeAVCeleb Khalid et al. (2021), and TalkingHeadBench Xiong et al. (2025), as well as our proposed RFAV dataset. We split the AVLips subset into training, validation, and test sets with a ratio of 6:1:3. Our RHTHFD is trained exclusively on real samples from the training set and evaluated on both the test set and other benchmark datasets. For FakeAVCeleb (FKAV), we construct the test set using 500 real samples along with 1,000 selected fake samples. For TalkingHeadBench (THB), the target test set is built by merging the official test sets of videos produced by all diffusion models. In the RFAV dataset, we use 2,000 videos covering different identities and forgery methods as the test set. As THB and RFAV lack real samples, we supplement their test sets with the real instances from the AVLips test split. For cross-manipulation generalization testing, we partition the test set based on the manipulation types defined in FKAV. Following prior work, we use Area Under Curve (AUC) and Average Precision (AP) as evaluation metrics.

**Implementation Details.** We compare unsupervised and supervised forgery detection methods. The unsupervised detectors include AVAD Feng et al. (2023), AVH-Align Smeu et al. (2025), AVBYOL Grill et al. (2020), and VQ-GAN Afouras et al. (2018a), while the supervised ones consist of CViT Wodajo & Atnafu (2021), LipFD Liu et al. (2024), and EfficientViT Coccomini et al. (2022). Our model is trained on real data, avoiding potential comparison bias caused by differences in forgery techniques. Moreover, most real training data for

Table 4: Average AWG weights for detecting different types of forged videos. Here, $w_1$ denotes the weight for local HSV feature stream, $w_2$ for global HSV feature stream, $w_3$ for audio features, and $w_4$ for fused visual features.

| Fake Videos | $w_1$ | $w_2$ | $w_3$ | $w_4$ |
|-------------|-------|-------|-------|-------|
| THB | 0.5876 | 0.4124 | 0.3575 | 0.6425 |
| RFAV | 0.5625 | 0.4375 | 0.3955 | 0.6045 |
| AVLips | 0.5558 | 0.4442 | 0.4624 | 0.5376 |

all methods are from real-world interview videos (YouTube or BBC), ensuring a fair comparison.

Dataset generation uses a single RTX 4090 GPU ( 8 GB memory) and requires 10 hours per identity for 3D reconstruction and metadata preparation, and approximately one GPU day per identity for real and forged video generation. Feature extraction with DinoV2 and a lip-reading network is performed offline. On an NVIDIA A100 GPU, processing 3,000 samples requires 6 GB memory and 30 minutes. Cached features allow inference of the same samples using 2 GB memory in 1.1 minutes. Detector training (excluding feature extraction) uses 4 GB memory on an A100 GPU and converges within 0.5 hours. *More results and details are in the Appendix A.*

## 5.2 EXPERIMENTAL RESULTS

**Cross-Dataset Generalization.** To evaluate the generalization performance of the proposed RHTHFD against different forgery methods, we conduct experiments on GAN-based datasets (AVLips and FKAV), a diffusion-based dataset (THB), and a radiance field-based dataset (RFAV). Our method and other unsupervised detectors learn solely from real videos and perform zero-shot fake video detection. As presented in Table 3, our HSV red hue-based model consistently surpasses both supervised and unsupervised baselines across all datasets. It is worth noting that RHTHFD is trained with substantially less data than competing methods.

Table 5: We report results on combinations of real/fake video/audio in the FKAV dataset composed of different manipulation algorithms. All methods are unsupervised and trained only on real samples. The best results are in **bold** and the second-best results are underlined.

| Methods | Modality | Training Videos | RVFA AP | RVFA AUC | FVRA-WL AP | FVRA-WL AUC | FVFA-FS AP | FVFA-FS AUC | FVFA-GAN AP | FVFA-GAN AUC | FVFA-WL AP | FVFA-WL AUC |
|---------|----------|-----------------|---------|----------|------------|-------------|------------|-------------|-------------|--------------|------------|-------------|
| AVBYOL | AV | 500k+ | 50.0 | 50.0 | 73.4 | 61.3 | 88.7 | 80.8 | 60.2 | 33.8 | 73.2 | 61.0 |
| VQ-GAN | V | 95k+ | - | - | 50.3 | 49.3 | 57.5 | 53.0 | 49.6 | 48.0 | 62.4 | 56.9 |
| AVAD | AV | 95k+ | 62.4 | 71.6 | 93.6 | 93.7 | 95.3 | 95.8 | 94.1 | 94.3 | 93.8 | 94.1 |
| AVAD | AV | 120k+ | 70.7 | 80.5 | 91.1 | 93.0 | 91.0 | 92.3 | 91.6 | 92.7 | 91.4 | 93.1 |
| RHTHFD (Ours) | AV | 2k | **99.7** | **98.9** | **99.4** | **98.0** | **98.8** | **96.6** | **98.7** | **96.1** | **98.9** | **96.7** |
| RHTHFD* (Ours) | AV | 1k | **99.7** | 98.6 | 99.3 | 97.3 | 98.3 | 94.3 | 98.2 | 93.6 | 98.4 | 94.3 |

On the GAN dataset, RHTHFD achieves the highest AP and AUC scores, with the performance improvement being even more pronounced on the AVLips. Compared to LipFD, a supervised method proposed alongside AVLips, RHTHFD improves AP and AUC by 10.3% and 11.2%, respectively. This is because GAN-generated faces show noticeable information loss in HSV space, making them easier to detect. On our proposed RFAV dataset, most existing methods perform poorly because radiance field-based talking head generation typically alters only mouth and eye movements, leaving other regions and head motion unaffected. In contrast, our method leverages discriminative features in the HSV space, achieving superior performance. In the THB dataset, diffusion-generated forgeries exhibit exceptional visual realism and audio-visual consistency, rendering most detection methods that rely on lip alignment and visual quality ineffective (AP and AUC both

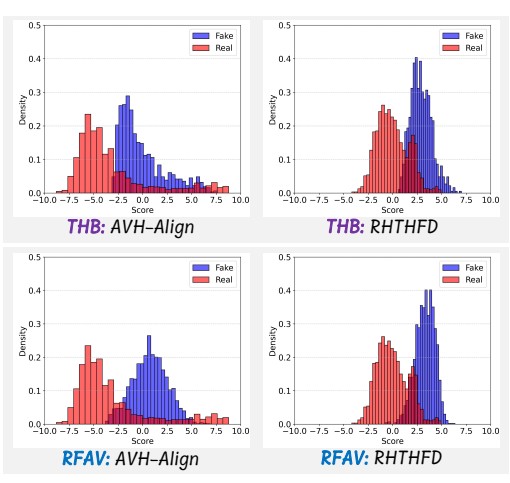

Figure 5: Histograms of score divergence between our method and AVH-Align on the RFAV and THB datasets.

below 50%). While AVH-Align demonstrates some discriminative capability, it remains insufficient. In Figure 1, diffusion methods tend to produce overly detailed facial features, which our method effectively exploits to achieve best performance. When trained with only half of the data, RHTHFD* still outperforms other methods across nearly all datasets, demonstrating the robustness of our method.

To better demonstrate the effectiveness of our method, Figure 5 presents score distribution histograms. On both the RFAV and THB datasets, our RHTHFD clearly distinguishes real samples from fake ones, with score distributions that are smaller and more concentrated. In contrast, AVH-Align exhibits a wider score range and includes stubborn samples that cannot be detected. These results strongly confirm that HSV features are highly effective in detecting fake videos generated by different forgery methods.

Furthermore, we quantitatively analyze how RHTHFD allocates attention when detecting forged videos. We focus on its response to local and global HSV feature streams, as well as to fused visual and speech representations. As shown in Table 4, we obtain the average weights from the AWG module for fake videos generated by diffusions (THB), radiance fields (RFAV), and GANs (AVLips). For the most challenging diffusion-based forgeries, the model focuses more on local HSV features and fused visual representations. A similar pattern is observed for radiance field-based forgeries. In contrast, GAN-generated videos are easier to detect in the HSV space, requiring less reliance on local and visual features. This observation aligns with the HSV color distribution patterns shown in Figure 1. For instance, the differences between diffusion forgeries and real samples are primarily reflected in fine-grained local visual features.

**Cross-Manipulation Generalization.** We follow the standard protocol established in AVAD Feng et al. (2023) to evaluate the cross-manipulation generalization ability on the FKAV dataset. The FKAV dataset is divided into five categories based on the video editing methods: (1) RVFA, which contains real video paired with fake audio; (2) FVRA-WL, where Wav2Lip Prajwal et al. (2020) generates fake video synchronized with real audio; (3) FVFA-WL, consisting of fake video generated by Wav2Lip driven by fake audio; (4) FVFA-FS, which combines fake audio with FaceSwap Korshunova et al. (2017) and Wav2Lip to syn-

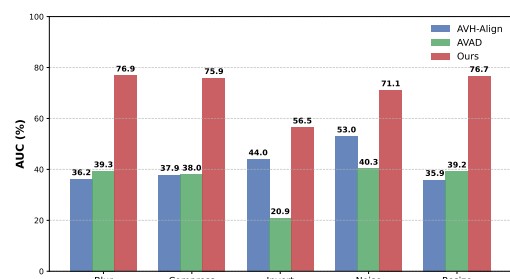

Figure 6: Robustness comparison under different perturbations on the THB dataset.

thesize fake video; and (5) FVFA-GAN, where fake audio is used with FaceSwapGAN Nirkin et al. (2019) and Wav2Lip to produce fake video. All fake audio samples are generated using SV2TTS Jia et al. (2018). This categorization allows for a systematic and comprehensive assessment of the ability to generalize across different types of audio-visual manipulations.

As shown in Table 5, our method achieves the best results across all five categories. VQ-GAN relies on a codebook to compress visual signals for forgery detection, so we do not report its performance on real videos. Remarkably, our method demonstrates highly consistent performance, achieving over 96% AP and AUC in most categories. Furthermore, even when trained on only half of the data, our detector still outperforms nearly all baselines. In the RVFA, our method shows a substantial improvement, primarily due to the mismatch between real video and fake audio, which our audio-visual consistency features effectively capture. For other settings, the performance gains are at-

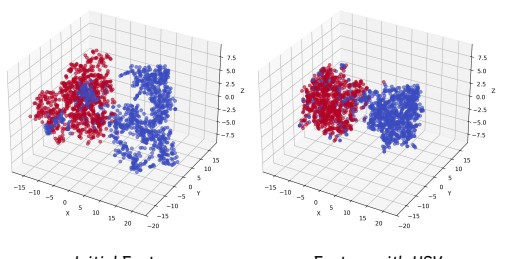

Figure 7: Visualization of visual features. *Initial Feature* are the raw visual features, while *Feature with HSV* include the HSV red hue enhancement.

tributed to the discriminative distributions of the red hue in HSV space and our carefully designed network architectures.

**Robustness Evaluation.** To evaluate robustness, we test model performance under various perturbations, including blur, compress, invert, noise, and resize. As shown in Figure 6, AVH and AVAD suffer significant performance degradation under these perturbations. Their AUC values mostly fall below 50%. In contrast, our method consistently outperforms these baselines across all perturbation types. These results highlight the effectiveness of leveraging the HSV red hue for audio-visual forgery detection.

**Visualization Results.** In Figure 7, when the HSV red hue feature is not introduced, the feature distributions of different classes are scattered and show substantial overlap, indicating that relying solely on raw visual features is insufficient for robust discrimination. After incorporating red hue enhancement, the feature distributions become much more compact, intra-class clustering is tighter, inter-class boundaries are clearer, and misclassified samples are significantly reduced. These results

confirm that the HSV red hue feature enables the model to more effectively capture discriminative cues, thereby improving overall detection performance and robustness.

### 5.3 ABLATION STUDY

To demonstrate the effectiveness of our contributions, we conduct ablation studies on different components, with the results presented in Table 6.

**Impact of HSV features.** When the distribution of the red hue is excluded, the model performance drops significantly on both datasets. This trend is most pronounced on the GAN-based AVLips, where AP decreases by 15.0% and AUC drops by 8.9%. This is because, compared to radiance field-based forgeries, GAN-generated videos exhibit more pronounced information loss in the HSV space. These results highlight the critical contribution of red hue distribution to audio-visual talking head deepfake detection.

**Analysis of Local and Global features.** Global and local HSV features are jointly used to enhance the fake detection capability. We perform ablation studies to assess the individual importance of each component. Removing the global features leads to a moderate performance drop, indicating that they provide foundational support but contribute less to capturing critical forgery cues. In contrast, excluding the local features results in a substantial decline in overall performance, suggesting that fine-grained artifacts captured by local features play a crucial role in forgery detection.

**Ablation of AWG module.** We use the AWG module to dynamically adjust the contribution of different features. When all features are fused with equal weighting, performance consistently declines across datasets. These results demonstrate the importance of adaptive weighting in forgery detection, particularly for identifying videos synthesized by diffusion and radiance field methods.

**Importance of Visual Region Attention.** We replace the visual region attention with a simple feature concatenation strategy to fuse HSV features and visual features. This configuration results in the poorest performance among all variants. The decline is primarily due to the disruption of spatial structure caused by concatenation, which hinders the model from focusing on discriminative HSV regions—particularly around the mouth. These results underscore the rationality and effectiveness of our model design.

**Effect of HSV Hue Selection.** In the HSV color space, hues can be categorized into red, green, and blue. The ablation results show that using the red hue consistently achieves the best performance across different test sets, followed by green and blue. This observation is consistent with the color intensity analysis in Figure 9 ($red > green > blue$), thereby validating the effectiveness of the red hue for forgery detection.

Table 6: Ablation study of the main components of RHTHFD on the AVLips and RFAV datasets, evaluated in terms of AP (%) and AUC (%). The best results are in **bold** and the second-best results are underlined.

| Methods | AVLips | | RFAV | |
|---|---|---|---|---|
| | AP | AUC | AP | AUC |
| w/o HSV Red Hue | 80.6 | 87.0 | 92.2 | 87.5 |
| w/o Local Feature | 85.0 | 88.7 | 92.3 | 85.5 |
| w/o Global Feature | 91.2 | 92.1 | 95.3 | 91.1 |
| w/o AWG module | 90.8 | 91.8 | 94.8 | 89.0 |
| w/o Visual Region Att. | 77.5 | 86.2 | 91.8 | 84.9 |
| w/ HSV Blue Hue | 93.2 | 93.9 | 94.1 | 93.7 |
| w/ HSV Green Hue | 93.8 | 94.2 | 94.5 | 93.9 |
| RHTHFD | **95.6** | **95.9** | **96.3** | **95.4** |

## 6 CONCLUSION

We have developed a novel HSV red hue-based feature for talking head deepfake detection, aiming to address the limitations of current detectors that overly rely on audio-visual alignment cues and struggle with highly realistic fake videos. Our analysis reveals significant differences in HSV color space between real and forged videos, which motivates the design of an unsupervised audio-visual detection model, RHTHFD. Additionally, we introduced the RFAV benchmark dataset, constructed using radiance fields, to overcome the limitations of existing baselines that rely solely on GANs or diffusion methods. Extensive experiments demonstrate that our method achieves remarkable performance across multiple datasets and different types of forgeries.

ETHICS STATEMENT

This work adheres to the ICLR Code of Ethics. Our research focuses on audio-visual talking head forgery detection, aiming to improve the security and reliability of multimedia communication. The dataset includes both publicly available resources and data collected from individuals. For the latter, explicit consent was obtained from the participant, and the data is used strictly for research purposes under agreed conditions. No private or sensitive information is disclosed.

REPRODUCIBILITY STATEMENT

We provide detailed experimental settings in Section 5.1 and Appendices A.1 and A.2. Results on mixed data and the impact of training data volume on model performance are reported in Appendix A.3. The key code used in our experiments is included in the supplementary material.

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

# A APPENDIX

## A.1 MORE DETAILS OF THE RFAV DATASET

### A.1.1 CONSTRUCTION METHODS

Existing audio-visual forgery detection datasets are typically constructed using GANs and diffusion models, overlooking radiance fields-based talking head generation methods. To address this gap, we

propose RFAV, a radiance field dataset built using Neural Radiance Fields (NeRF) Peng et al. (2024) and 3D Gaussian Splatting (3DGS) Cho et al. (2024). Unlike GANs and diffusion models that focus on 2D face synthesis, NeRF and 3DGS offer geometry-aware radiance field representations, enabling photorealistic talking head generation.

Table 7 presents a comparative analysis of eight radiance field-based talking head generation methods. We identify those that provide training code as candidate generators for the RFAV dataset. For reproducible models, AD-NeRF Guo et al. (2021) proposes an audio-driven talking head generation method that directly maps audio features to dynamic NeRF for portrait rendering. Geneface Ye et al. (2023) proposes a three-stage framework that enables NeRF-based talking face systems to leverage large-scale lip-reading corpora and achieve a certain level of generalization to various out-of-distribution audio inputs. ER-NeRF Li et al. (2023) introduces an efficient Tri-Plane Hash Representation to facilitate dynamic head reconstruction, while achieving real-time inference and fast convergence with a compact model size. SyncTalk Peng et al. (2024) proposes a system framework composed of a face-sync controller and a head-sync stabilizer, which effectively ensures smooth and synchronized head movements.

For 3DGS, TalkingGaussian Li et al. (2024a) introduces a deformation-based framework that synthesizes talking heads by applying deformations to a persistent head structure, thereby mitigating inherent facial distortions caused by inaccurate appearance predictions. GaussianTalker Cho et al. (2024) enhances spatial consistency among adjacent Gaussians by reformulating the 3D Gaussian representation into a feature volume representation. Additionally, it introduces cross-attention mechanisms between audio and spatial features to improve synthesis stability and enable region-specific deformation across a large number of Gaussians.

Table 7: Overview of talking head generation methods based on NeRF and 3DGS, with **bolded** entries indicating those used to construct RFAV dataset.

| Methods | Framework | Venu | Code Availability |
|---|---|---|---|
| AD-NeRF Guo et al. (2021) | NeRF | ICCV'21 | Yes |
| GeneFace Ye et al. (2023) | NeRF | ICLR'23 | Yes |
| ER-NeRF Li et al. (2023) | NeRF | ICCV'23 | Yes |
| **SyncTalk** Peng et al. (2024) | NeRF | CVPR'24 | Yes |
| PointTalk Xie et al. (2025) | 3DGS | AAAI'25 | No |
| GauTalker Yu et al. (2024) | 3DGS | ACM MM'24 | No |
| TalkingGaussian Li et al. (2024a) | 3DGS | ECCV'24 | Yes |
| **GaussianTalker** Cho et al. (2024) | 3DGS | ACM MM'24 | Yes |

Table 8: Quantitative and qualitative results of different radiance field-based methods. The best and second-best methods are highlighted in **bold** and underlined, respectively. ↑ indicates that higher values are better, while ↓ indicates that lower values are preferred.

| Methods | PSNR ↑ | LMD ↓ | LSE-C ↑ | Head Stability |
|---|---|---|---|---|
| AD-NeRF Guo et al. (2021) | 26.05 | 3.118 | 4.901 | noticeable jitter |
| GeneFace Ye et al. (2023) | 29.87 | 4.021 | 5.328 | noticeable jitter |
| ER-NeRF Li et al. (2023) | 31.22 | 2.989 | 5.720 | noticeable jitter |
| SyncTalk Peng et al. (2024) | **35.17** | **2.555** | **7.425** | **stable** |
| TalkingGaussian Li et al. (2024a) | 32.97 | 2.723 | 6.350 | noticeable jitter |
| GaussianTalker Cho et al. (2024) | 33.69 | 2.688 | 6.026 | slight jitter |

### A.1.2 GENERATION METHOD SELECTION

To select the final data synthesis method, we perform quantitative evaluations based on visual quality (PSNR), motion accuracy (LMD), and lip-sync consistency (LSE-C). Following the protocol of Li et al. (2024a), we train the target talking head using the training sets of four portrait video sequences from Ye et al. (2023); Peng et al. (2024), and evaluate the model performance on the corresponding

test sets of each identity. Additionally, we conduct manual observations to identify potential issues such as head jitter, which are often difficult to capture through quantitative metrics alone. For the evaluation of head motion stability, we categorize the results into three levels: stable, slight jitter, and noticeable jitter.

The results are reported in Table 8. It can be observed that SyncTalk achieves the best performance across all evaluation metrics. Although GaussianTalker performs slightly worse than TalkingGaussian in terms of lip-sync accuracy, it demonstrates advantages in other aspects and exhibits less head jitter. Therefore, we ultimately select SyncTalk and GaussianTalker as the generators for the RFAV dataset. In the RFAV dataset, 4,000 samples are synthesized using SyncTalk and 2,000 samples using GaussianTalker, with the driving audio sourced from Speech-forensics Ji et al. (2024). Figure 8 presents a subset of the synthesized results, demonstrating the high realism of the videos generated by the selected methods. This highlights the necessity of constructing a radiance field-based audio-visual forgery detection dataset.

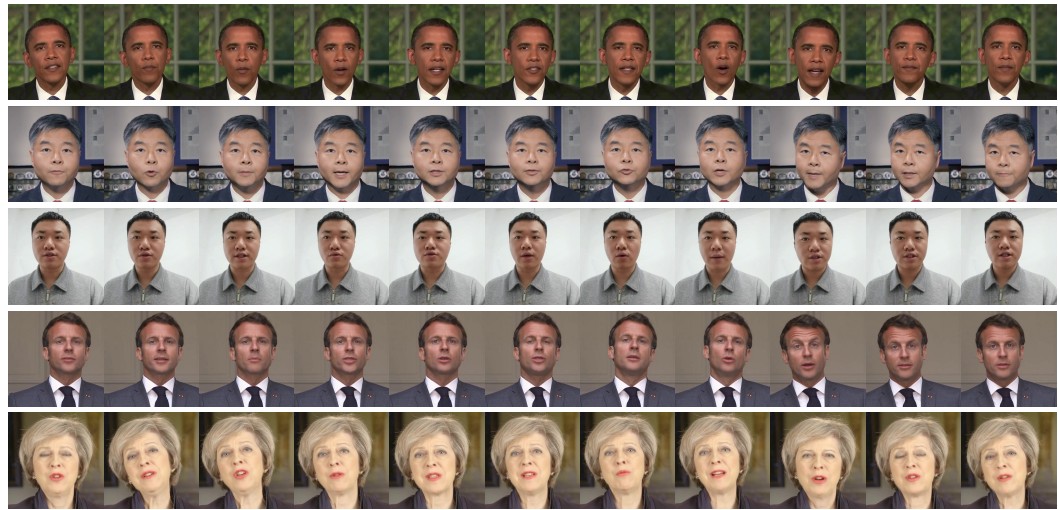

Figure 8: Examples of synthetic data. Please zoom in for better visualization.

## A.2 MORE EXPERIMENTAL DETAILS

### A.2.1 COLOR INTENSITY

As shown in Figure 9, we select 20 forged videos generated by different methods along with real videos, and compute the RGB color intensity frame by frame. It can be observed that the red channel consistently exhibits the highest intensity across all frames. Based on this observation, we perform hue mapping of the red in the HSV color space. To intuitively compare the distribution differences of red hue in HSV space between different forgery methods and real samples, we visualize the results for the same identity in Figure 10. Compared to the real talking head, videos generated by diffusion models show overly complex facial details due to excessive transition optimization, GAN-based methods struggle to reconstruct fine facial features, and radiance field-based methods lack sufficient facial detail reconstruction. These findings motivate us to reconsider talking head deepfakes from the perspective of red hue discrepancies in the HSV color space.

### A.2.2 ADAPTIVE WEIGHT GENERATOR

In RHTHFD, the Adaptive Weight Generator (AWG) module is designed to adaptively learn the weights of input features. The AWG module dynamically assigns weights to both local-global visual features and the fused visual-audio features, enabling the model to focus on more discriminative and distinctive features. The detailed architecture of AWG is illustrated in Figure 11. Specifically, features $a$ and $b$ are first concatenated, and then passed through a sequence of linear layers followed by ReLU activation functions. Finally, a softmax function is applied to generate weight values.

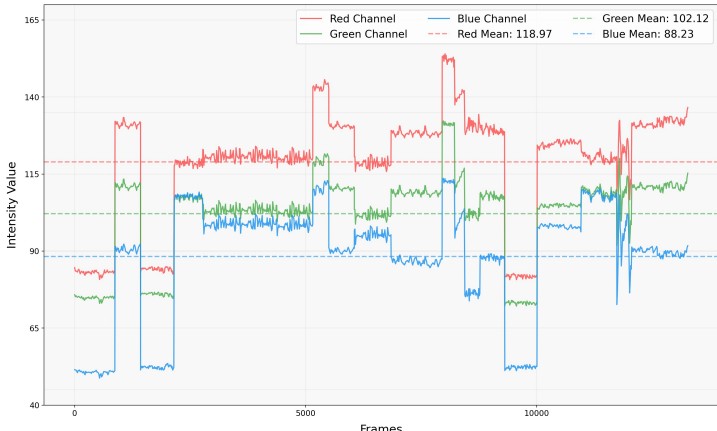

Figure 9: Color channel intensity across different frames. Please zoom in for better visualization.

### A.2.3 DETECTORS

In this work, we adopt supervised and unsupervised forgery detection methods as baselines. The unsupervised methods trained solely on real samples include AVAD Feng et al. (2023), AVH-Align Smeu et al. (2025), AVBYOL Grill et al. (2020), and VQ-GAN Afouras et al. (2018a). The supervised models trained on both real and fake samples include CViT Wodajo & Atnafu (2021), EfficientViT Coccomini et al. (2022), and LipFD Liu et al. (2024).

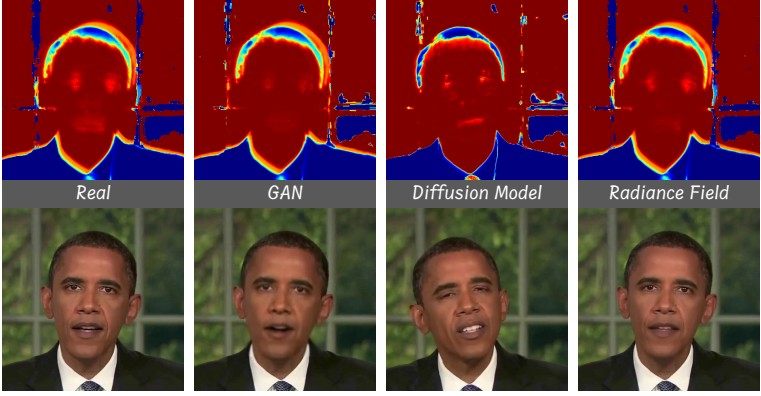

Figure 10: Red hue mapping in the HSV color space for real and forged videos of the same identity. The forgery methods include GAN-based Wang et al. (2023), diffusion-based Zheng et al. (2024), and radiance field-based Cho et al. (2024) generators.

AVAD is an audio-visual forgery detection method. It first estimates the desynchronization between audio and video, and then determines whether these patterns are typical of real data or indicative of anomalies. AVH-Align identifies that the presence of leading silence in forged videos introduces bias in deepfake detection. To mitigate this issue, it proposes an unsupervised method that avoids learning such dataset-specific artifacts, thereby improving the robustness of the model. AVBYOL is an audio-visual contrastive learning model that determines whether the visual and audio streams in a video are synchronized. VQ-GAN is a method trained on a large-scale audio-visual dataset for anomaly detection. Consistent with Feng et al. (2023), we use the log-likelihood of the codebook averaged over each video frame to perform anomaly detection.

As for the supervised methods, CViT is a vision-based forgery detection method that learns local and global image features by combining convolutional neural networks with the attention mechanism of the Transformer architecture. EfficientViT combines multiple types of vision transformers with a convolutional EfficientNet B0 as the feature extractor, and has achieved promising progress in video

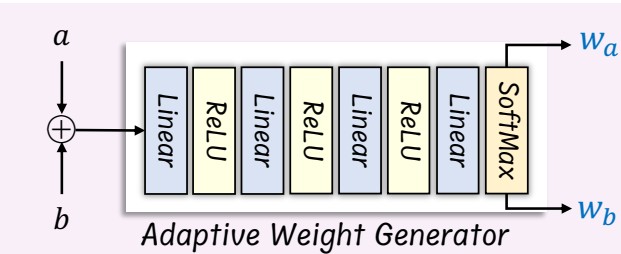

Figure 11: Detailed architecture of the Adaptive Weight Generator module.

forgery detection tasks. LipFD is a supervised audio-visual talking head forgery detection method that leverages discrepancies between audio and lip movements to identify forged videos.

### A.2.4 PUBLIC BENCHMARKS

In addition to our proposed radiance field-based RFAV dataset, we conduct experiments on FakeAVCeleb (FKAV) Khalid et al. (2021) and AVLips Liu et al. (2024), which are constructed using GAN-based generators, as well as on TalkingHeadBench (THB) Xiong et al. (2025), a dataset synthesized using diffusion models. This comprehensive evaluation allows us to assess the generalization performance of detection models across different forgery generation techniques.

The FKAV dataset comprises 500 real videos sourced from VoxCeleb2 Chung et al. (2018) and 19.5k forged videos. The fake samples are generated using various techniques, including Faceswap Korshunova et al. (2017), FSGAN Nirkin et al. (2019), and Wav2Lip Prajwal et al. (2020), as well as audio forgeries synthesized via SV2TTS Jia et al. (2018). We utilize FKAV to conduct both cross-dataset and cross-manipulation generalization experiments. For the cross-dataset evaluation, to mitigate the imbalance between real and fake samples, we construct a test set consisting of 500 real videos (independent from the training sets of all methods) and 1,000 fake videos generated using diverse forgery techniques. For the cross-manipulation generalization experiment, we adopt the evaluation protocol from Feng et al. (2023), categorizing videos based on the type of forgery method. Each editing category includes 100 real videos and 500 forged videos, enabling a fine-grained analysis of model robustness across different manipulation types.

AVLips is an audio-visual lip-sync dataset specifically constructed for talking head forgery detection. It contains approximately 340,000 audio-video samples generated using various lip-sync methods. To simulate realistic lip movements, the dataset integrates both static generation techniques such as MakeItTalk Zhou et al. (2020) and dynamic methods including Wav2Lip, TalkLip Wang et al. (2023), and SadTalker Zhang et al. (2023). The real samples are primarily sourced from LRS3 Afouras et al. (2018b), FaceForensics++ Rossler et al. (2019), DFDC Dolhansky et al. (2020), and real-world data. On the official AVLips website, the authors do not provide access to the full dataset. The available subset includes real samples from LRS3 and synthetic videos generated using unspecified GAN-based methods. We split this subset into training, validation, and test sets with a ratio of 6:1:3. Both our method and the cutting-edge unsupervised AVH-Align are trained on 2,000 real samples from the training set.

THB is a recently proposed talking head forgery detection dataset primarily constructed using diffusion-based methods. It synthesizes forged videos by combining portrait images from FFHQ Karras et al. (2019) with driving signals from CelebV-HQ Zhu et al. (2022). The dataset includes samples generated by six representative methods: Hallo Xu et al. (2024), Hallo2 Cui et al. (2024), AniPortrait (Audio-driven, AniAudio), AniPortrait (Video-driven, AniVideo) Wei et al. (2024), LivePortrait Guo et al. (2024), and EMOPortraits Drobyshev et al. (2024), covering both state-of-the-art diffusion and GAN-based generators. Additionally, THB incorporates commercial data generated by the diffusion-based MAGI-1 Sand-AI (2025). The official THB repository provides separate training, validation, and test sets for each generation method. For our evaluation, we aggregate all test samples generated by diffusion-based methods into a unified target test set used in our paper. Specifically, the test set includes 117 videos synthesized by Hallo, 105 by Hallo2, 146 by AniPortrait (Audio-driven), 108 by AniPortrait (Video-driven), and 66 commercial samples generated by MAGI-1.

Table 9: Corruption settings applied to video frames for robustness evaluation. Each perturbation simulates common real-world degradations.

| Corruption Type | Parameter | Range | Value |
|---|---|---|---|
| JPEG Compression (`compress`) | quality | 0–100 | 20 |
| Resize (`resize`) | scale | 0–1 | 0.5 |
| Gaussian Blur (`blur`) | ksize | odd integer | 5 |
| Gaussian Noise (`noise`) | $\sigma$ | $\geq 0$ | 25 |
| Color Inversion (`invert`) | — | N/A | N/A |

### A.2.5 Robustness Evaluation

Due to the vulnerability of in-the-wild videos to various corruptions, a good detector must not only demonstrate strong generalization but also withstand common perturbations to accurately identify forged content. Therefore, we study the performance of the detector under five types of perturbations. JPEG compression introduces detail loss and compression artifacts, simulating video encoding degradation (lower quality, stronger artifacts). Resizing reduces resolution and blurs fine-grained structures (smaller scale, more blur). Gaussian blur smooths high-frequency content, potentially obscuring subtle facial and lip movements (larger kernel, stronger blur). Gaussian noise simulates sensor or transmission errors, testing robustness to random pixel-level variations (larger $\sigma$, stronger noise). Color inversion alters visual appearance while preserving spatial structure, assessing the ability to handle extreme color shifts (inverts pixel values). The settings of each perturbation are summarized in Table 9. The visual results under different perturbations are shown in Figure 12.

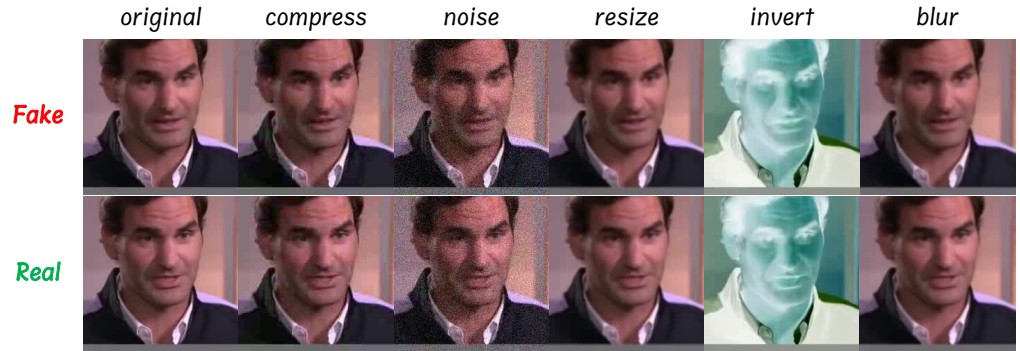

Figure 12: Real and fake videos are processed with common perturbation methods, and video frames are then extracted to obtain samples.

### A.3 More Experimental Results

#### A.3.1 Results on Mixed Data

To evaluate model performance under adversarial conditions that closely resemble complex real-world scenarios, we construct a mixed test set by combining videos generated from three heterogeneous synthesis paradigms (GANs, diffusion, radiance fields). Specifically, the real samples in this test set are consistent with those in the RFAV dataset, while the fake samples are composed of forged videos from RFAV, AVLips, and THB. As shown in Table 10, our method consistently achieves the best performance even under more complex forgery scenarios. Furthermore, when the training data is reduced by half, our method still significantly outperforms other baselines, demonstrating its effectiveness and strong generalization capability.

#### A.3.2 Training Data Volume

We found that RHTHFD consistently achieves optimal results across various experimental settings, and this advantage remains significant even when the training data is halved. To investigate the

Table 10: Results on the mixed data. We report AP (%) and AUC (%) with the best results in **bold** and the second-best results underlined. sup. and unsup. refer to supervised and unsupervised methods. RHTHFD* indicates a model trained on half of the RHTHFD dataset.

| Methods | Type | AUC | AP |
|---|---|---|---|
| CViT Wodajo & Atnafu (2021) | sup. | 86.4 | 68.8 |
| EfficientViT Coccomini et al. (2022) | sup. | 81.7 | 57.3 |
| LipFD Liu et al. (2024) | sup. | 82.1 | 62.3 |
| AVAD Feng et al. (2023) | unsup. | 88.2 | 69.8 |
| AVH-Align Smeu et al. (2025) | unsup. | 88.6 | 84.1 |
| RHTHFD (Ours) | unsup. | **98.2** | **95.1** |
| RHTHFD* (Ours) | unsup. | 97.0 | 92.7 |

impact of training data volume on model performance, we conducted experiments using mixed test data. As shown in Figure 13, even with a very limited training set (only 100 videos), our method still outperforms supervised approaches such as LipFD and EfficientViT. As the training data increases to around 700 videos, RHTHFD surpasses the latest unsupervised method AVH-Align. This trend highlights the importance and effectiveness of our design choices—specifically, the incorporation of the red hue from the HSV color space and the tailored network architecture—for audio-visual forgery detection.

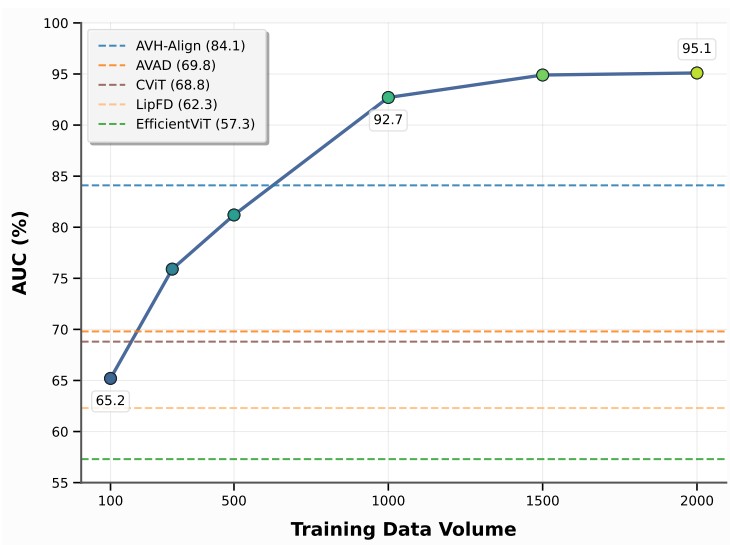

Figure 13: Impact of training data volume on model performance. All methods are evaluated on a test set that integrates forgery data generated by GANs, diffusion models, and neural radiance fields.

### A.3.3 RESULTS ON MORE PERTURBATIONS

We also evaluate RHTHFD on THB under five common perturbations: AAC, Brightness, Gamma, H264 compression, and White Balance. Results in Table 11 show that RHTHFD remains highly stable, demonstrating strong robustness to color and compression changes. This stability arises because red hue features capture subtle, locally structured forgery traces that are minimally affected by global color or brightness shifts. In contrast, AVH-Align shows performance gains under these perturbations, indicating sensitivity to post-processing and limited stability. AVAD exhibits little variation but consistently underperforms, suggesting weak robustness. Even under stronger perturbations (Figure 6), while all methods degrade, RHTHFD maintains the highest performance, confirming that red hue features capture intrinsic and resilient forgery signals.

Table 11: Performance under more perturbations. We report AP (%) / AUC (%) for each method.

| Perturbation | RHTHFD (AP / AUC) | AVH-Align (AP / AUC) | AVAD (AP / AUC) |
|---|---|---|---|
| AAC | 89.5 / 93.5 | 81.6 / 87.2 | 43.7 / 48.0 |
| Brightness | 87.5 / 92.1 | 81.8 / 86.6 | 43.5 / 46.7 |
| Gamma | 90.6 / 93.3 | 84.4 / 87.9 | 43.6 / 47.3 |
| H264 | 90.5 / 93.7 | 85.2 / 88.1 | 43.7 / 47.7 |
| White Balance | 89.2 / 92.9 | 65.3 / 82.7 | 43.7 / 47.7 |
| Original (No Perturbation) | 89.1 / 93.0 | 64.8 / 82.3 | 43.8 / 48.1 |

### A.3.4 THE RATIONALE FOR USING DINOV2

DinoV2 is pretrained on large-scale RGB natural images, and its Transformer architecture with self-attention effectively captures spatial textures and local patterns. In talking head fake detection, the HSV red hue encodes subtle and locally structured forgery traces. Applying DinoV2 to HSV allows us to extract rich local and global features while focusing on robust red hue anomalies. Experimental results on THB in Table 12 show that DinoV2-HSV significantly outperforms DinoV2-RGB, demonstrating the advantage of combining DinoV2 with HSV red-hue features. We also attempt to replace DinoV2 with DinoV3 as the HSV feature extractor. The experimental results show that DinoV3 performs slightly better than DinoV2, indicating some advantage in capturing red hue anomalies. However, the parameter size of DinoV3 is about 6 times larger than that of DinoV2, which leads to a significant increase in computational cost. Considering both the performance gain and the model complexity, we still recommend using DinoV2, as it provides sufficiently stable HSV feature representations. This also demonstrates that our red hue feature itself offers strong discriminative capacity and does not merely rely on backbone upgrades.

Table 12: Comparison of different visual feature backbones. We report AP (%) and AUC (%) for each method, with the best results in **bold** and the second-best results underlined.

| Method | AP (%) | AUC (%) |
|---|---|---|
| DinoV2-RGB | 82.7 | 89.9 |
| DinoV3-HSV | **90.2** | **93.9** |
| DinoV2-HSV | 89.1 | 93.0 |

### A.3.5 THE RATIONALE FOR FUSING AUDIO FEATURES

The task of talking head fake detection inherently relies on both audio and visual modalities. While the core contribution of our method lies in enhancing visual features through HSV red hue, the overall performance of the model largely relies on audio-visual consistency. When we remove the audio modality, the model almost completely fails, indicating that enhanced visual features alone are insufficient for robust forgery detection. Moreover, our base audio-visual features are extracted using a lip-reading network, which encodes the correspondence between speech and facial movements and thus provides cross-modal synchronization information. Therefore, removing the audio modality breaks this speech–visual consistency constraint and deprives the model of critical discriminative cues.

### A.3.6 OUT-OF-DOMAIN COMPARISONS

We conduct tests on a new singing scenario, collecting 400 real singing videos and 600 forged videos. It is important to note that this scenario differs significantly from our original training data (talking head videos) in terms of content, recording conditions, and performance style, and thus represents a typical out-of-domain evaluation. The experimental results in Table 13 show that, despite the clear distributional gap from the training data, our RHTHFD still achieves around 70% in AP and AUC, demonstrating substantial discriminative ability compared to random guessing, while other methods completely fail in this scenario. This indicates that even under audio-visual conditions with large distributional differences from the training data, the red hue can still effectively capture

discriminative features. In the future, we will explore cross-scenario generalization to build a more universal audio-visual forgery detection method.

Table 13: Out-of-domain evaluation on a new scenario. AP (%) and AUC (%) are reported for each method, with the best results in **bold**.

| Method | AP (%) | AUC (%) |
|---|---|---|
| AVAD | 50.4 | 47.4 |
| AVH-Align | 35.3 | 20.4 |
| **RHTHFD** | **70.2** | **69.4** |

### A.3.7 ANALYSIS OF ADVERSARIAL VULNERABILITIES

To verify the adversarial vulnerabilities of the red hue feature in enhancing talking head deepfake detection, we construct "distribution-matched" forged videos by adjusting the red hue histogram of each forged video to match the distribution of real videos. The results on THB dataset in Table 14 show that even under this adversarial setting, the model performance remains stable. This indicates that our model does not rely solely on global red hue statistics, but instead leverages the red hue feature to capture more essential and locally structured facial anomalies. Even when the overall red hue distribution is matched to real videos, these features still provide discriminative capacity. These findings confirm the important role of the red hue feature in robust detection.

Table 14: Evaluation of adversarial robustness. AP (%) and AUC (%) are reported for the original features and features matched on the red hue.

| Method | AP (%) | AUC (%) |
|---|---|---|
| Red-Hue Matched | 88.6 | 92.5 |
| Original | 89.1 | 93.0 |

### A.3.8 VISUALIZATION OF THE EFFECTIVENESS OF HSV FEATURES

As shown in Figure 14. The initial visual features (blue) exhibit high standard deviations, which often obscure local anomalies with noise. After introducing HSV red hue enhancement (red), the overall standard deviation narrows and the feature distribution becomes more stable, enabling the model to capture consistent class information. A few frames still show sharp deviations, corresponding to extreme forgery artifacts or lighting variations, indicating that HSV features suppress global noise while retaining sensitivity to anomalous frames. In terms of the mean, the initial features fluctuate considerably, whereas the enhanced features produce a smoother curve that highlights stable class differences. Overall, HSV features strengthen model robustness while preserving the ability to discriminate local forgery signals.

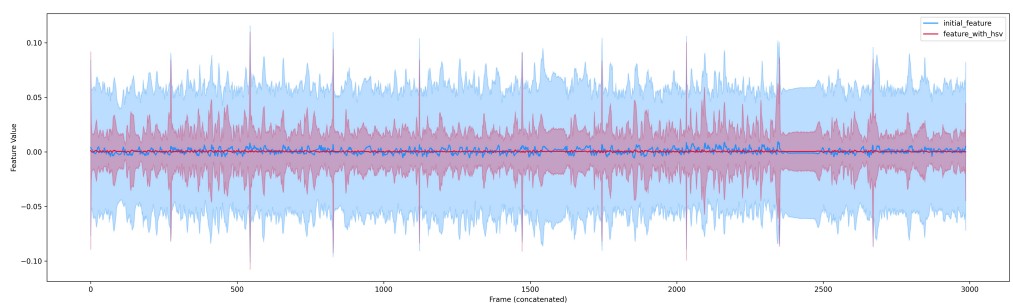

Figure 14: Visualization of mean and standard deviation distributions. The red and blue solid lines indicate the mean values, and the vertical axis represents the standard deviation range.

### A.3.9 OTHER COLOR SPACES

In Table 15, we compare the HSV red hue feature with YCbCr and Lab color spaces on THB. When replacing the feature with Cb/Cr (YCbCr) or a*/b* (Lab), detection performance dropped significantly, whereas HSV maintained substantially higher performance. This indicates that the differences we exploit are not a general phenomenon that can be captured by any color space. The reason is that the hue component in HSV better decouples color from luminance and highlights red variations closely related to physiological properties of skin, which remain difficult for current talking head forgery methods to faithfully reconstruct. Therefore, the HSV red hue feature provides robust and discriminative forgery cues that cannot be substituted by other color spaces.

Table 15: Comparison of different color spaces. AP (%) and AUC (%) are reported for each color space, with the best results in **bold**.

| Color Space | AP (%) | AUC (%) |
|---|---|---|
| YCbCr | 82.1 | 89.7 |
| Lab | 81.4 | 88.3 |
| **HSV** | **89.1** | **93.0** |

### A.3.10 SENSITIVITY TO RESOLUTION, LIGHTING, AND SKIN QUALITY

To evaluate the robustness of our method under variations in resolution, lighting, and skin quality, we conduct systematic perturbation experiments. Specifically, we test standard and extreme downsampling (resize 0.5 and 0.25) to assess resolution effects, apply gamma correction with factors 1.2, 0.6, and 1.5 to simulate lighting variations, and add Gaussian noise ($\sigma$=25 and $\sigma$=30) to mimic coarse or rough skin textures. The results in Table 16 indicate that our method maintains strong detection performance across all perturbations. Even under extreme resolution reduction or significant lighting changes, the red hue remains effective with only moderate performance degradation. While performance decreases slightly under simulated poor skin quality, our model generally outperforms baseline detectors, confirming the robustness of the physiologically grounded red hue features. These findings demonstrate that the proposed method is resilient to realistic video acquisition variations, supporting its applicability for challenging real-world talking head forgery detection scenarios.

Table 16: Robustness evaluation under resolution, lighting, and skin quality perturbations. AP (%) and AUC (%) are reported for each method, with the best results in **bold**.

| Perturbation | RHTHFD (AP / AUC %) | AVH-Align (AP / AUC %) | AVAD (AP / AUC %) |
|---|---|---|---|
| Resize 0.5 | **74.2 / 76.7** | 37.2 / 35.9 | 39.2 / 39.2 |
| Resize 0.25 | **72.8 / 73.2** | 36.9 / 34.2 | 37.5 / 38.8 |
| Gamma 1.2 | **90.6 / 93.3** | 84.4 / 87.9 | 43.6 / 47.3 |
| Gamma 0.6 | **87.6 / 92.0** | 79.1 / 80.6 | 43.5 / 46.6 |
| Gamma 1.5 | **89.4 / 92.4** | 82.6 / 86.3 | 43.1 / 46.4 |
| Noise ($\sigma$=25) | **65.2 / 71.1** | 46.3 / 53.0 | 40.1 / 40.3 |
| Noise ($\sigma$=30) | **63.9 / 69.1** | 40.4 / 44.7 | 35.5 / 36.9 |
| Original | **89.1 / 93.0** | 64.8 / 82.3 | 43.8 / 48.1 |

### A.3.11 DATASET QUALITY ANALYSIS

**Identity Pool.** The primary goal of RFAV is not to serve as a large-scale training dataset, but rather as a high-quality evaluation benchmark specifically designed to address the lack of radiance field talking head synthesis in existing forgery datasets. Due to the high production cost of radiance-field–driven talking head synthesis, each identity requires independent modeling and optimization, which inherently limits the total number of identities. Importantly, each identity in RFAV is associated with a large number of real and synthetic speech-driven videos, encompassing diverse facial

dynamics and lip movements. Therefore, despite the limited identity pool, RFAV sufficiently captures the intra-identity variability necessary to evaluate talking head forgery detection methods under this novel generative paradigm.

**Gender and Language Imbalance.** The primary goal of RFAV is to provide a representative evaluation benchmark that includes identities across different genders, languages, and age groups. Even with a limited number of identities, the dataset is still valuable for analyzing model robustness across key demographic variables, which is crucial for assessing talking head forgery detection methods.

**Coverage of Skin Tones.** Although the RFAV dataset does not fully cover all possible skin tones, it includes Asian, White, and Black identities, representing the typical range encountered in talking head forgery detection research.

**Quality Assessment.** We conduct a systematic quality evaluation using an equal number of videos from each generative paradigm, computing metrics such as Flow Magnitude (FM Mean/FM Std) to capture motion naturalness and stability, Temporal Flicker (TF Mean/TF Std) to measure frame-to-frame brightness inconsistency, Laplacian Variance (LV) to evaluate image detail richness, Frame-Wise Smoothness (FWS) to quantify facial landmark motion smoothness, Landmark Missing Rate (LMR) to assess landmark detection failure, and LSE-C to evaluate audio-visual lip-sync consistency.

In Table 17, the Radiance Field samples consistently exhibit lower flicker and smoother frame-to-frame transitions than the other paradigms, indicating more stable rendering. Their optical flow lies between the overly smooth motion of Diffusion and the excessive dynamics of GAN, reflecting a balanced and realistic motion profile. In terms of visual detail, Radiance Field videos achieve LV scores comparable to Diffusion and higher than GAN, confirming rich image structures. Regarding landmark quality, RFAV also performs strongly, with FWS competitive with Diffusion and LMR reaching 0.0000, matching the best-performing paradigm. Additionally, Radiance Field videos demonstrate excellent lip-sync alignment, with LSE-C scores substantially higher than Diffusion and comparable to GAN. Overall, these results demonstrate that Radiance Field–generated data possess consistently high temporal and perceptual fidelity, combining stable motion, minimal flicker, sharp appearance, reliable facial geometry, and accurate speech-driven synchronization. This level of quality confirms that RFAV provides a robust and coherent benchmark suitable for evaluating talking head forgery detection models under a generative paradigm that has been absent from prior datasets.

Table 17: Data quality evaluation across different generative paradigms.

| Paradigm | FM Mean | FM Std | TF Mean | TF Std | LV | FWS | LMR | LSE-C |
|---|---|---|---|---|---|---|---|---|
| Diffusion | 0.3961 | 0.2755 | 2.2116 | 1.1077 | 136.16 | 0.0079 | 0.0000 | 4.258 |
| GAN | 0.9658 | 0.6306 | 4.3729 | 2.4049 | 131.84 | 0.0208 | 0.0104 | 7.301 |
| Radiance Field | 0.5081 | 0.3365 | 1.9001 | 0.9390 | 135.17 | 0.0100 | 0.0000 | 7.246 |

## A.4 DISCUSSION AND FUTURE WORK

Most existing audio-visual forgery detection datasets primarily focus on single-speaker manipulations. They rarely address scenarios involving multiple speakers, such as cases where only one speaker is forged while others remain authentic. This limitation reduces the generalization capability of current detection methods that rely heavily on audio-visual synchronization. We believe that our proposed RHTHFD has the potential to handle such complex settings. This is because our method leverages the distribution of red hue in the HSV color space for forgery detection, a feature that remains applicable even in multi-speaker environments.

Furthermore, current audio-visual forgery detection methods, including our proposed RHTHFD, are typically designed for identifying whether a video is real or fake. They do not possess the ability to trace the origin of the forgery, such as determining which specific generation algorithm or manipulation technique was used. This limitation reduces their effectiveness in forensic applications, where understanding the source of the forgery is essential for accountability, legal investigation, and the development of targeted defense strategies.

We plan to advance our research in the following two directions:

- **Development of a Multi-Speaker Audio-Visual Forgery Detection Dataset.** We aim to construct a dataset tailored for multi-speaker environments. Investigating the challenges and variations in lip movements across different individuals will contribute to building more robust and accurate detection models for talking head deepfakes.

- **Development of Forgery Attribution Methods.** We aim to explore methods that go beyond identifying whether content is real or fake, and enable attribution of forged content to specific generation algorithms or manipulation techniques. By analyzing subtle artifacts and model-specific patterns embedded in the audio-visual data, we seek to build models capable of identifying the origin of the forgery.

### A.5 USE OF LARGE LANGUAGE MODELS (LLMs)

We used large language models (LLMs) solely as a general-purpose assistive tool for language polishing and grammar checking. The initial draft of the manuscript, as well as all ideas, experiments, results, and analyses, were developed entirely by the authors without any LLMs' assistance. LLMs were not involved in research design, data processing, or scientific reasoning, and their use did not influence the technical content or conclusions of the work.

