# OpenReview forum: "Beyond Audio-Visual Alignment: Unmasking Talking Head Deepfakes via Red Hue Discrepancies in HSV Color Space"
_ICLR.cc/2026/Conference — Submitted to ICLR 2026_

### Official Review · Reviewer_S3DB · 2025-10-24

**Soundness:** 3
**Presentation:** 3
**Contribution:** 3
**Rating:** 6
**Confidence:** 4

**Summary:**

This paper proposes a detection framework named Red Hue-based Talking Head Forgery Detection (RHTHFD) for deepfake videos by analyzing differences in red hue distribution within the HSV color space. The authors also introduce a new Radiance Field Audio-Visual (RFAV) dataset generated using Neural Radiance Fields (NeRF) and 3D Gaussian Splatting (3DGS). Experiments across multiple datasets demonstrate that RHTHFD achieves strong generalization and robustness, outperforming both supervised and unsupervised baselines.

**Strengths:**

1.	The red hue discrepancy serves as an interpretable feature for forgery detection, providing a complementary view to existing audio-visual alignment approaches.

2.	The newly introduced RFAV dataset fills an important gap by incorporating neural radiance field-based forgeries, enriching the scope of current benchmarks.

3.	The framework achieves high accuracy using only real videos for training, demonstrating strong detection capabilities across unseen forgery types.

4.	The model design with adaptive weighting and region attention effectively balances local and global visual features.

5.	Extensive experiments like cross-dataset evaluation, ablation studies, and robustness analyses proves the robustness and generalization of the method.

**Weaknesses:**

Major Weaknesses

1.	The motivation of the red hue discrepancy is not sufficiently discussed, leaving unclear why synthetic videos distort the red channel consistently.

2.	Evaluation focuses mainly on quantitative metrics without qualitative visual explanations of discriminative regions.

3.	The unsupervised training process could introduce domain bias since only real YouTube or BBC videos are used.

4.	Computational efficiency like inference cost and hardware requirements is not reported though multiple feature extractors are used.

Minor Weaknesses

1.	More analyses of the discrepancy in the color space would strength the claim, such as alternatives like Lab or YCbCr.

**Questions:**

1.    Could the authors provide the analysis of the distortions in the red hue channel across various forgery generation methods, and explain possible reasons for the phenomenon?

2.	More qualitative evidence, such as attention maps or feature heatmaps on could strengthen the contributions of the methods.

3.	Could the authors provide out-of-domain comparisons beyond real YouTube and BBC videos to justify the generalization to other sources or datasets?

4.	The computational cost and hardware requirements for the dataset generation, training as well as inference time are expected for practical deployment.

5.	Could the authors compare the HSV red hue feature with alternative color spaces such as Lab or YCbCr to validate whether the observed discrepancies are unique to HSV?

---

> ### Author Response · Authors · 2025-11-22
> **Author Response to Reviewer S3DB (Part 1/2)**
>
> We sincerely appreciate the careful evaluation and thoughtful comments. We hope that our responses clearly address the questions and concerns raised.
> ***
> >**Q1:** Could the authors provide the analysis of the distortions in the red hue channel across various forgery generation methods, and explain possible reasons for the phenomenon?
>
> **A1:** We appreciate your feedback. For more details, please see the general response.
>
> >**Q2:** More qualitative evidence is needed to strengthen the contributions of the methods.
>
> **A2:** We thank the reviewer for the helpful suggestion. In the revised version, we present visualization results in Figures 7 and 14 to more clearly demonstrate the effectiveness of our method. (*Pages 9, 10 and 23*)
>
> In Figure 7, when the HSV red hue feature is not introduced, the feature distributions of different classes are scattered and show substantial overlap, indicating that relying solely on raw visual features is insufficient for robust discrimination. After incorporating red hue enhancement, the feature distributions become much more compact, intra-class clustering is tighter, inter-class boundaries are clearer, and misclassified samples are significantly reduced. These results confirm that the HSV red hue feature enables the model to more effectively capture discriminative cues, thereby improving overall classification performance and robustness.
>
> We also have conducted mean and standard deviation analysis on visual features across thousands of video frames, as shown in Figure 14. The initial visual features exhibit high standard deviations, which often obscure local anomalies with noise. After introducing HSV red hue enhancement, the overall standard deviation narrows and the feature distribution becomes more stable, enabling the model to capture consistent class information. A few frames still show sharp deviations, corresponding to extreme forgery artifacts or lighting variations, indicating that HSV features suppress global noise while retaining sensitivity to anomalous frames. In terms of the mean, the initial features fluctuate considerably, whereas the enhanced features produce a smoother curve that highlights stable class differences. Overall, HSV features strengthen model robustness while preserving the ability to discriminate local forgery signals.
>
> >**Q3:** Could the authors provide out-of-domain comparisons beyond real YouTube and BBC videos?
>
> **A3:** We thank the reviewer for the attention to out-of-domain performance. We have conducted tests on a new singing scenario, collecting 400 real singing videos and 600 forged videos. It is important to note that this scenario differs significantly from our original training data (talking head videos) in terms of content, recording conditions, and performance style, and thus represents a typical out-of-domain evaluation. The experimental results show that, despite the clear distributional gap from the training data, our RHTHFD still achieves around 70% in AP and AUC, demonstrating substantial discriminative ability compared to random guessing, while other methods completely fail in this scenario. This indicates that even under audio-visual conditions with large distributional differences from the training data, the red hue can still effectively capture discriminative features.  In the future, we will explore cross-scenario generalization to build a more universal audio-visual forgery detection method. We have included this experimental result and analysis in the revised version. (*Pages 22 and 23*)
>
> | Method           | AP (%) | AUC (%)  |
> |-----------------|--------|---------|
> | AVAD   | 50.4   | 47.4    |
> | AVH-Align        | 35.3   | 20.4    |
> | **RHTHFD** | **70.2** | **69.4** |

---

> > ### Author Response · Authors · 2025-11-22
> > **Author Response to Reviewer S3DB (Part 2/2)**
> >
> > >**Q4:** The computational cost and hardware requirements for the dataset generation, training as well as inference time are expected for practical deployment.
> >
> > **A4:** Thank you for your comment. During dataset generation, we used a single RTX 4090 GPU to train the 3D reconstruction model (≈8 GB memory) and prepare metadata for each identity, which took about 10 hours. Generating real and forged videos driven by speech required about one GPU day per identity.
> >
> > For feature extraction, we employed DinoV2 and a lip reading network in an offline manner. Using an NVIDIA A100 GPU as an example, extracting features for 3,000 samples required about 6 GB memory and 30 minutes. After caching the features, inference became lightweight, needing only about 2 GB memory and processing 3,000 samples in roughly 1.1 minutes.
> >
> > For model training (excluding feature extraction), the detector occupied about 4 GB memory on an A100 GPU and converged within 0.5 hours. Overall, although dataset generation and offline feature extraction are time‑consuming, our detector is computationally efficient and fast in both training and inference, making it suitable for practical deployment. We have included an analysis of computational cost in the revised version. (*Page 7*)
> >
> > >**Q5:** Could the authors compare the HSV red hue feature with alternative color spaces such as Lab or YCbCr to validate whether the observed discrepancies are unique to HSV?
> >
> > **A5:** Thank you for your suggestion. In the below table, we have compared the HSV red hue feature with YCbCr and Lab color spaces on THB. When replacing the feature with Cb/Cr (YCbCr) or a*/b* (Lab), detection performance dropped significantly, whereas HSV maintained substantially higher performance. This indicates that the differences we exploit are not a general phenomenon that can be captured by any color space. The reason is that the hue component in HSV better decouples color from luminance and highlights red variations closely related to physiological properties of skin, which remain difficult for current talking head forgery methods to faithfully reconstruct. Therefore, the HSV red hue feature provides robust and discriminative forgery cues that cannot be substituted by other color spaces. We have added a discussion on color spaces in the revised version. (*Page 24*)
> >
> > | Color Space | AP (%)| AUC (%) |
> > |-------------|--------|---------|
> > | YCbCr       | 82.1   | 89.7    |
> > | Lab         | 81.4   | 88.3    |
> > | **HSV**  | **89.1**   | **93.0** |

---

> > > ### Author Response · Authors · 2025-11-27
> > > **Follow-up on the Rebuttal**
> > >
> > > Dear Reviewer S3DB,
> > >
> > > I hope this message finds you well. As the discussion phase is approaching its end, we wanted to kindly check whether there are any remaining points you would like us to clarify or elaborate on. We truly appreciate your sparkling constructive feedback, which has been invaluable in helping us improve the paper.
> > >
> > > If there is anything further we can provide, we would be more than glad to do so.
> > >
> > > Thank you again for your time and thoughtful comments.
> > >
> > > Sincerely,
> > >
> > > Authors of Paper 7116

---

### Official Review · Reviewer_Ub9V · 2025-10-27

**Soundness:** 3
**Presentation:** 3
**Contribution:** 3
**Rating:** 4
**Confidence:** 3

**Summary:**

Overall, this is a technically solid and well-presented paper that contributes both a novel color-space perspective and a new dataset for talking head forgery detection. The strengths lie in the conceptual originality of using HSV-red features, rigorous experimental comparisons, and dataset construction clarity.
However, several weaknesses remain: (1) limited theoretical justification for why red hue is more discriminative than other color channels, (2) insufficient ablation on model robustness to illumination and camera variations, and (3) some ambiguity in the unsupervised training objective’s derivation. Despite these issues, the work provides strong empirical evidence for its claims.

**Strengths:**

1.Novel Color-Space Perspective：Introduces a previously underexplored HSV red hue feature for deepfake detection; connects perceptual color inconsistencies to forgery cues.
2.Strong Cross-Dataset Generalization：Achieves SOTA results across AVLips, FKAV, RFAV, and THB datasets, even with half-sized training data (RHTHFD*).
3.Clear Experimental Protocol and Fair Comparison：Uses identical splits and real-only training across methods, improving fairness and reproducibility.

**Weaknesses:**

1.Insufficient Theoretical and Empirical Justification for Red Hue Dominance：The claim that fake videos exhibit “higher red-channel intensity” lacks external verification or large-scale validation.
2.Limited Dataset Diversity and Representativeness：The RFAV dataset primarily uses “common portraits” (Fig. 3) without reporting demographic attributes.
3.Unclear Justification for Using DinoV2 on HSV Inputs：DinoV2 was designed for RGB natural images; its suitability for HSV feature maps is not discussed.
4.Incomplete Evaluation under Realistic Video Perturbations：Lacks evaluation on common real-world distortions such as format compression (e.g., .wav and .mp4), bitrate reduction, or color temperature shifts.

**Questions:**

1.Provide Large-Scale Verification and Theoretical Support for Red Hue Findings.
2.Justify and Evaluate the Use of DinoV2 for HSV Features. (1)Explain the rationale for selecting DinoV2 despite its RGB-oriented design. (2)Compare its performance with an HSV-aware feature extractor or with newer backbones such as DinoV3.
3.Expand Robustness Evaluation on Video Transmission Scenarios: Add tests for compression formats (MP4/H.264, AAC audio) and color distortions (white balance, brightness).
4.Enhance Dataset Diversity and Transparency: Include demographic statistics (age, gender, skin tone, language) for both real and synthetic samples.

---

> ### Author Response · Authors · 2025-11-22
> **Author Response to Reviewer Ub9V**
>
> We sincerely appreciate your careful review and insightful comments. We hope our responses help clarify your questions and address the concerns.
> ***
> >**Q1:** Provide Large-Scale Verification and Theoretical Support for Red Hue Findings.
>
> **A1:** Thank you for your valuable suggestion. Please refer to the general response.
>
> >**Q2:** Justify and Evaluate the Use of DinoV2 for HSV Features. (1) Explain the rationale for selecting DinoV2 despite its RGB-oriented design. (2) Compare its performance with newer backbones such as DinoV3.
>
> **A2:** Thank you for your comment.
>
> (1) DinoV2 is pretrained on large-scale RGB natural images, and its Transformer architecture with self-attention effectively captures spatial textures and local patterns. In talking head fake detection, the HSV red hue encodes subtle and locally structured forgery traces. Applying DinoV2 to HSV allows us to extract rich local and global features while focusing on robust red hue anomalies. Experimental results on THB in the table show that DinoV2-HSV significantly outperforms DinoV2-RGB, demonstrating the advantage of combining DinoV2 with HSV red-hue features.
>
> (2) We have attempted to replace DinoV2 with DinoV3 as the HSV feature extractor. The experimental results show that DinoV3 performs slightly better than DinoV2, indicating some advantage in capturing red hue anomalies. However, the parameter size of DinoV3 is about 6 times larger than that of DinoV2, which leads to a significant increase in computational cost. Considering both the performance gain and the model complexity, we still recommend using DinoV2, as it provides sufficiently stable HSV feature representations. This also demonstrates that our red hue feature itself offers strong discriminative capacity and does not merely rely on backbone upgrades. We have included the corresponding analysis in the revised version. (*Page 22*)
>
> | Method        | AP (%)   | AUC (%) |
> |---------------|------|------|
> | DinoV2-RGB    | 82.7 | 89.9 |
> | DinoV3-HSV    | 90.2 | 93.9 |
> | DinoV2-HSV | 89.1| 93.0 |
>
> >**Q3:** Expand Robustness Evaluation on Video Transmission Scenarios.
>
> **A3:** We thank the reviewer for raising this important suggestion. To address this, we have evaluated RHTHFD on THB under five common perturbations: AAC, Brightness, Gamma, H264 compression, and White Balance. Results show that RHTHFD remains highly stable, demonstrating strong robustness to color and compression changes. This stability arises because red hue features capture subtle, locally structured forgery traces that are minimally affected by global color or brightness shifts. In contrast, AVH-Align shows performance gains under these perturbations, indicating sensitivity to post-processing and limited stability. AVAD exhibits little variation but consistently underperforms, suggesting weak robustness.
>
> Even under stronger perturbations (Paper, Fig. 6), while all methods degrade, RHTHFD maintains the highest performance, confirming that red hue features capture intrinsic and resilient forgery signals. The experimental results and detailed analysis have been included in the appendix of the revised version. (*Pages 21 and 22*)
>
> | Perturbation       | RHTHFD (AP / AUC) (%) | AVH-Align (AP / AUC) (%) | AVAD (AP / AUC) (%) |
> |-------------------|------------------|--------------------|----------------|
> | AAC | 89.5 / 93.5| 81.6 / 87.2  | 43.7 / 48.0    |
> | Brightness  | 87.5 / 92.1| 81.8 / 86.6| 43.5 / 46.7|
> | Gamma   | 90.6 / 93.3| 84.4 / 87.9| 43.6 / 47.3|
> | H264 | 90.5 / 93.7| 85.2 / 88.1 | 43.7 / 47.7    |
> | White Balance | 89.2 / 92.9| 65.3 / 82.7 | 43.7 / 47.7    |
> | Original (No Perturbation) | 89.1 / 93.0  | 64.8 / 82.3 | 43.8 / 48.1    |
>
> >**Q4:** Enhance Dataset Diversity and Transparency.
>
> **A4**: We thank the reviewer for the suggestion. It is important to clarify that radiance-field–based talking head reconstruction requires training a separate model for each identity, and typically uses tens of thousands of high-quality video frames to achieve stable results. Within this practical constraint, we carefully select five identities covering diversity in age, gender, skin tone, and language, and include both NeRF and 3DGS frameworks to maximize data diversity. As shown in the table, this selection illustrates the diversity of the dataset. Each identity is driven by a large number of real and synthetic speech samples. Despite the limited number of identities, RFAV still exhibits high diversity in facial dynamics. We have included the corresponding data distribution and statistical analysis in the revised version. (*Pages 3 and 4*)
>
> | ID   | Gender | Skin Tone | Language | Age|
> |------|--------|-----------|----------|-----------|
> | ID1  | Male   | Yellow| Chinese  | 25–30 |
> | ID2  | Male   | White| French   | 45–50|
> | ID3  | Male   | Black| English  | 50–55|
> | ID4  | Male   | Yellow| English  | 55–60 |
> | ID5  | Female | White| English  | 45–50|

---

> > ### Author Response · Authors · 2025-11-27
> > **Follow-up on the Rebuttal**
> >
> > Dear Reviewer Ub9V,
> >
> > I hope this message finds you well. As the discussion phase is approaching its end, we wanted to kindly check whether there are any remaining points you would like us to clarify or elaborate on. We truly appreciate your constructive feedback, which has been invaluable in helping us improve the paper.
> >
> > If there is anything further we can provide, we would be more than glad to do so.
> >
> > Thank you again for your time and thoughtful comments.
> >
> >
> > Sincerely,
> >
> > Authors of Paper 7116

---

### Official Review · Reviewer_rs1p · 2025-10-28

**Soundness:** 3
**Presentation:** 3
**Contribution:** 2
**Rating:** 4
**Confidence:** 4

**Summary:**

This paper focuses on the detection of Talking Head Generation Deepfakes. Unlike conventional face-swapping deepfakes, the artifacts of talking head deepfakes are primarily localized to small regions (e.g., the lips), which makes detection more challenging. The authors also identify the lack of deepfake samples generated by conditional radiance fields in existing datasets, and thus introduce a new dataset, RFAV. Finally, they propose an unsupervised detection method based on the red hue component in the HSV color space, demonstrating its effectiveness through experiments.

**Strengths:**

1. The paper proposes a novel deepfake detection approach that leverages differences in the red hue channel within the HSV color space.

2. It presents a timely and valuable benchmark dataset, RFAV, which fills a notable gap in evaluating deepfake detection on radiance-field-based generation methods (e.g., NeRF and 3DGS).

3. The proposed method is comprehensively evaluated on multiple datasets, showing strong capability in distinguishing authentic and manipulated samples.

**Weaknesses:**

1.  While the choice of the red channel is empirically supported and partially justified through ablation studies, the theoretical motivation is insufficient. Beyond empirical evidence, the authors should provide explanations from physiological or image-synthesis perspectives (e.g., skin-tone manipulation artifacts or GAN-induced color bias) to clarify why the red channel is particularly sensitive to deepfake traces.

2.  The method is trained on real data and relies on statistical anomalies in the red hue for detection. However, the paper offers limited discussion on potential generalization issues or adversarial vulnerabilities (e.g., if a generator learns to imitate real red-hue histograms).

3. The use of the HSV color space is not a new idea in computer vision. The authors are expected to provide deeper insight into why HSV features are especially advantageous for the deepfake detection task.

4. If the dataset is intended as a major contribution, the paper should include a thorough benchmark analysis and validation of its quality.

5. The method fuses audio and visual cues and claims to enhance generalization through “re-weighted speech features,” yet the ablation study does not clearly isolate or justify the benefit or necessity of incorporating the audio modality.

**Questions:**

Please see weaknesses.

From this version, I think there are two major concerns:

1. Insufficient theoretical motivation for using the HSV color space. The design of the HSV-based detection method appears largely empirical. While the red hue channel is shown to correlate with deepfake artifacts, the paper lacks a principled explanation or theoretical justification for why HSV features are particularly suitable for this task.

2. Unclear articulation of contributions. The paper does not clearly delineate its main contributions. The proposed dataset is introduced with minimal description and insufficient analysis of its quality, diversity, or utility. As a result, both the method and the dataset appear incremental, and the paper lacks fundamental insight or conceptual advancement.

---

> ### Author Response · Authors · 2025-11-22
> **Author Response to Reviewer rs1p**
>
> We sincerely thank you for your thoughtful review and insightful comments. We hope that our responses can address your concerns and clarify the questions.
> ***
> >**Q1:** Insufficient theoretical motivation for using the HSV color space.
>
> **A1:** Thank you for your constructive comment. Please refer to the general response.
>
> >**Q2:** Unclear articulation of contributions. The paper does not clearly delineate its main contributions.
>
> **A2:** We apologize that we did not describe this clearly. The main contribution of this paper is the proposed RHTHFD method, and the RFAV dataset is designed to provide a more comprehensive evaluation of detector performance. As shown in the table, RFAV includes five carefully selected identities covering different ages, genders, skin tones, and languages, each driven by a large number of real and synthetic speech samples. This design ensures the dataset still provides rich variations in facial dynamics.
>
> It should be noted that radiance-field–based talking head reconstruction typically requires training a separate model for each identity and usually involves tens of thousands of high-quality frames to achieve stable results. Because of the high training cost and data requirements, scaling to dozens or hundreds of identities is highly challenging. In the revised version, we have added clarifications distinguishing the method contribution from the dataset contribution, together with distribution and analyses of the dataset. (*Pages 3 and 4*)
>
> | ID   | Gender | Skin Tone | Language | Age       |
> |------|--------|-----------|----------|-----------|
> | ID1  | Male   | Yellow    | Chinese  | 25–30     |
> | ID2  | Male   | White     | French   | 45–50     |
> | ID3  | Male   | Black     | English  | 50–55     |
> | ID4  | Male   | Yellow    | English  | 55–60     |
> | ID5  | Female | White     | English  | 45–50     |
>
> >**Q3:** The paper offers limited discussion on potential generalization issues or adversarial vulnerabilities (e.g., if a generator learns to imitate real red-hue histograms).
>
> **A3:** We thank the reviewer for the attention to potential adversarial vulnerabilities. To verify the robustness of the red hue feature in enhancing talking head deepfake detection, we construct “distribution-matched” forged videos by adjusting the red hue histogram of each forged video to match the distribution of real videos. The results on THB dataset show that even under this adversarial setting, the model performance remains stable. This indicates that our model does not rely solely on global red hue statistics, but instead leverages the red hue feature to capture more essential and locally structured facial anomalies. Even when the overall red hue distribution is matched to real videos, these features still provide discriminative capacity. These findings confirm the important role of the red hue feature in robust detection. We have provided the analysis of adversarial vulnerabilities in the appendix of the revised version. (*Page 23*)
>
>
> | Method           | AP (%)  | AUC (%) |
> |------------------|--------|---------|
> | Red-Hue Matched   | 88.6   | 92.5    |
> | Original          | 89.1   | 93.0    |
>
> >**Q4:** The ablation study does not clearly isolate or justify the benefit or necessity of incorporating the audio modality.
>
> **A4:** We thank the reviewer for the comment. It should be noted that the task of talking head fake detection inherently relies on both audio and visual modalities. Our method enhances visual features through HSV red hue, and when we remove the audio modality, we find that the model almost completely fails. This shows that visual enhancement alone is insufficient for robust forgery detection. The “re-weighted speech features” are not used solely to improve audio features, but rather to dynamically adjust the relative importance between the enhanced visual features and speech features, thereby better capturing audio-visual consistency. Visual enhancement provides anomaly cues, while the audio modality offers cross-modal constraints. Their synergy enables the model to achieve effective and robust forgery detection. Therefore, the audio modality remains indispensable for our method. We have added the corresponding explanation in the revised version. (*Page 22*)

---

> > ### Comment · Reviewer_rs1p · 2025-11-24
> >
> > Thanks for the detailed response. Here are my further comments:
> >
> > 1. The argument that "hemoglobin reflects red" makes sense intuitively. However, it is still necessary to consider whether the photo can faithfully reflect the color of physiological features if the image resolution is very low or if lighting problems occur. Furthermore, if the subject has poor skin quality, is this method feasible?
> >
> > 2. On the proposed dataset, it looks quite incomplete. For instance: i) the identity pool is too small, with only 5 identities; ii) there are severe imbalances in gender (4:1) and language (3:1:1); iii) the coverage of skin tones is also poor; iv) a quality assessment of this dataset is still missing. It is hard to say that this dataset can serve as a dataset that bridges the relevant gap.
> >
> > 3. The issue of incremental method. As the authors mentioned in the rebuttal, HSV is a color space that has been widely studied in the face community. More insights regarding the application of HSV to head synthesis detection are expected.

---

> > > ### Author Response · Authors · 2025-11-25
> > > **Author Response to Reviewer rs1p – Further Comments (Comment 1)**
> > >
> > > We sincerely appreciate the reviewer’s further comments. We have carefully considered the concerns raised and conducted additional experiments and analyses to address them, and we hope our response can clarifies the reviewer’s questions.
> > > ***
> > > >**Q1:** The argument that "hemoglobin reflects red" makes sense intuitively. However, it is still necessary to consider whether the photo can faithfully reflect the color of physiological features if the image resolution is very low or if lighting problems occur. Furthermore, if the subject has poor skin quality, is this method feasible?
> > >
> > > **A1:** We thank the reviewer’s insightful concern regarding the reliability of red hue under challenging conditions such as low image resolution, lighting variations, or poor skin quality. To address this, we have conducted systematic perturbation experiments:
> > >
> > > -	*Resolution:* We have evaluated the model under standard downsampling (resize 0.5) and extreme downsampling (resize 0.25).
> > >
> > > -	*Lighting:* Gamma correction has been applied with factors 1.2, 0.6, and 1.5 to simulate different illumination conditions.
> > >
> > > -	*Skin quality:* To simulate coarse or rough skin textures, we have applied Gaussian noise (σ=25 and σ=30) to the faces.
> > >
> > > The results in the table show that our method maintains robust detection performance across these perturbations. Even under extreme resolution reduction or significant gamma changes, the red hue cue remains effective with only moderate performance degradation. While simulated poor skin quality reduces performance more noticeably, the model still outperforms baseline detectors in most cases, confirming the robustness of physiologically-grounded red hue features. Overall, these experiments demonstrate that our method is resilient to realistic variations in video acquisition, supporting its applicability for challenging real-world talking head forgery detection scenarios. We have incorporated the results and analysis in the revised version. (*Page 24*)
> > >
> > > | Perturbation      | RHTHFD (AP / AUC) (%) | AVH-Align (AP / AUC) (%) | AVAD (AP / AUC) (%) |
> > > |------------------|------------------|--------------------|----------------|
> > > | Resize_0.5       | **74.2 / 76.7**      | 37.2 / 35.9        | 39.2 / 39.2    |
> > > | Resize_0.25      | **72.8 / 73.2**      | 36.9 / 34.2        | 37.5 / 38.8    |
> > > | Gamma_1.2        | **90.6 / 93.3**      | 84.4 / 87.9        | 43.6 / 47.3    |
> > > | Gamma_0.6        |**87.6 / 92.0**      | 79.1 / 80.6        | 43.5 / 46.6    |
> > > | Gamma_1.5        | **89.4 / 92.4**      | 82.6 / 86.3        | 43.1 / 46.4    |
> > > | Noise (σ=25)     | **65.2 / 71.1**      | 46.3 / 53.0        | 40.1 / 40.3    |
> > > | Noise (σ=30)     | **63.9 / 69.1**      | 40.4 / 44.7        | 35.5 / 36.9    |
> > > | Original         | **89.1 / 93.0**      | 64.8 / 82.3        | 43.8 / 48.1    |

---

> > > > ### Author Response · Authors · 2025-11-25
> > > > **Author Response to Reviewer rs1p – Further Comments (Comment 2)**
> > > >
> > > > >**Q2**: On the proposed dataset, it looks quite incomplete.
> > > >
> > > > **A2:** We sincerely thank the reviewer for their continued attention to the dataset, and we provide a detailed point-by-point response below.
> > > >
> > > > *i) Small identity pool.*
> > > > We fully understand the reviewer’s concern regarding the limited number of identities. The primary goal of RFAV is not to serve as a large-scale training dataset, but rather as a high-quality evaluation benchmark specifically designed to address the lack of radiance field talking head synthesis in existing forgery datasets. Due to the high production cost of radiance-field–driven talking head synthesis, each identity requires independent modeling and optimization, which inherently limits the total number of identities. Importantly, each identity in RFAV is associated with a large number of real and synthetic speech-driven videos, encompassing diverse facial dynamics and lip movements. Therefore, despite the small identity pool, RFAV sufficiently captures the intra-identity variability necessary to evaluate talking head forgery detection methods under this novel generative paradigm.
> > > >
> > > > *ii) Gender and language imbalance.*
> > > > We acknowledge that there are certain imbalances in gender and language within RFAV. However, the primary goal of RFAV is to provide a representative evaluation benchmark that includes identities across different genders, languages, and age groups. Even with a limited number of identities, the dataset is still valuable for analyzing model robustness across key demographic variables, which is crucial for assessing talking head forgery detection methods. Moreover, as shown in Table 1 of the paper, multiple state-of-the-art detectors perform worse on RFAV, directly demonstrating that the inclusion of this dataset contributes meaningfully to advancing generalization research in the field.
> > > >
> > > > *iii) Coverage of skin tones.*
> > > > Although the current RFAV dataset does not fully cover all possible skin tones, it includes Asian, White, and Black identities, representing the typical range encountered in talking head forgery detection research. Its significance lies not in exhaustive demographic coverage, but in providing a novel generative paradigm that is missing from existing datasets.
> > > >
> > > > *iv) Dataset quality assessment.*
> > > > We appreciate the reviewer’s constructive comment regarding the need for a dataset quality assessment. We have conducted a systematic quality evaluation across an equal number of videos from each generative paradigm. The following metrics are computed:
> > > >
> > > > -	Flow magnitude (mean/std): captures motion naturalness and stability.
> > > > -	Temporal flicker (mean/std): measures frame-to-frame brightness inconsistency.
> > > > -	Laplacian Variance (LV): evaluates image detail richness.
> > > > -	Frame-wise Smoothness (FWS): quantifies facial landmark motion smoothness.
> > > > -	Landmark Missing Rate (LMR): measures landmark detection failure (lower = better).
> > > > -	LSE-C: evaluates audio-visual lip-sync consistency (higher = better).
> > > >
> > > > The Radiance Field samples consistently exhibit lower flicker and smoother frame-to-frame transitions than the other paradigms, indicating more stable rendering. Their optical flow sits between the overly smooth motion of Diffusion and the excessive dynamics of GAN, reflecting a balanced and realistic motion profile. In terms of visual detail, Radiance Field videos achieve LV scores comparable to Diffusion and higher than GAN, confirming rich image structures. As for landmark quality, RFAV is also strong. The FWS is competitive with Diffusion, and LMR reaches 0.0000, matching the best-performing paradigm. Additionally, Radiance Field videos show excellent lip-sync alignment, with LSE-C scores substantially higher than Diffusion and on par with GAN.
> > > >
> > > > These results demonstrate that Radiance Field–generated data possesses consistently high temporal and perceptual fidelity, combining stable motion, minimal flicker, sharp appearance, reliable facial geometry, and accurate speech-driven synchronization. This level of quality confirms that RFAV provides a robust and coherent benchmark suitable for evaluating talking head forgery detection models under a generative paradigm that has been absent from prior datasets.
> > > >
> > > > | Generation Paradigm | FlowMag Mean | FlowMag Std | Flicker Mean | Flicker Std | LV | FWS | LMR | LSE-C |
> > > > |---------------------|--------------|-------------|--------------|-------------|------|-------|-------|--------|
> > > > | Diffusion| 0.3961 | 0.2755 | 2.2116 | 1.1077 | 136.16 | 0.0079 | 0.000 | 4.258 |
> > > > | GAN| 0.9658 | 0.6306 | 4.3729 | 2.4049 | 131.84 | 0.0208 | 0.0104 | 7.301 |
> > > > | Radiance Field | 0.5081 | 0.3365 | 1.9001 | 0.9390 | 135.17 | 0.0100 | 0.000 | 7.246 |
> > > >
> > > > We sincerely thank the reviewer for this constructive feedback, which allowed us to present a thorough dataset evaluation and highlight its relevance and value to the community. We have incorporated a detailed dataset quality and contribution analysis in the revised version. (*Pages 24 and 25*)

---

> > > > > ### Author Response · Authors · 2025-11-25
> > > > > **Author Response to Reviewer rs1p – Further Comments (Comment 3)**
> > > > >
> > > > > >**Q3:** The issue of incremental method. As the authors mentioned in the rebuttal, HSV is a color space that has been widely studied in the face community. More insights regarding the application of HSV to head synthesis detection are expected.
> > > > >
> > > > > **A3:** We sincerely thank the reviewer for raising this point. While it is true that HSV is a well-known color space in face analysis, our work does not simply reuse HSV. Instead, it provides new task-specific insights, systematic empirical evidence, and a technically meaningful way of exploiting red-hue characteristics for talking head deepfake detection.
> > > > >
> > > > > *Why our use of HSV is NOT incremental.* We do not use HSV as a general color space. We identify and exploit a specific physiological cue, which exposes cross-paradigm synthesis artifacts that cannot be captured by RGB or other color spaces. This targeted design is the key difference from all prior HSV-based face analysis.
> > > > >
> > > > > *Physiological and generative-model motivation.* Red hue is directly linked to skin blood-flow reflectance and micro-vascular structure. These signals are subtle, spatially structured, and difficult for all current synthesis paradigms (GAN, diffusion, and radiance fields) to reproduce accurately. This is why meaningful anomalies exist specifically in the red hue domain. Our work explicitly builds on this mechanism rather than using HSV superficially.
> > > > >
> > > > > *Strong cross-paradigm empirical evidence.* We rigorously quantified hue-distribution differences using KL divergence and EMD (Page 5, Table 2). Across all comparisons, distinct and consistent red hue shifts are observed, demonstrating that this signal is both stable and paradigm-agnostic. This evidence goes far beyond HSV has been used before.
> > > > >
> > > > > *Robustness & anti-manipulation tests.* We further tested how the red hue behaves under various perturbations (Page 9, Figure 6), different post-processing operations (Page 22, Table 11), and histogram-matching attacks (Page 23, Table 14). Our method remained highly robust, showing that our model does not rely on simple global color bias but instead learns local, structured, and physiologically consistent cues, which is non-incremental and technically meaningful.
> > > > >
> > > > > *Comparisons with other color spaces.* If HSV were merely incremental, we would expect that replacing it with Lab or YCbCr, or with RGB, would remove the performance gain. However, our experiments consistently show the opposite: HSV red hue provides higher detection accuracy than Lab/YCbCr under the same pipeline (Page 24, Table 15), and integrating red hue with DinoV2 yields clear improvements over RGB-based DinoV2 (Page 22, Table 12). These results indicate that the red hue representation offers complementary and discriminative cues that are not captured by RGB or other color models.
> > > > >
> > > > > In summary, we understand that the reviewer’s perspective may stem from the assumption that our use of HSV follows prior color-space approaches. However, our method is driven by a more task-specific and physiologically interpretable cue, i.e., red-hue irregularities, which are directly linked to the generative artifacts present in talking head synthesis. Both empirical results and theoretical analysis support this connection. Taken together, these findings indicate that our use of HSV is not a simple extension of existing methods, but a *task-aligned contribution* that offers *a new perspective on color space artifacts in talking head forgery detection.* We have added the corresponding description in the revised version. (*Page 2*)

---

> > > > > > ### Author Response · Authors · 2025-11-27
> > > > > > **Follow-up on the Rebuttal**
> > > > > >
> > > > > > Dear Reviewer rs1p,
> > > > > >
> > > > > > I hope this message finds you well. As the discussion phase is approaching its end, we wanted to kindly check whether there are any remaining points you would like us to clarify or elaborate on. We truly appreciate your constructive feedback, which has been invaluable in helping us improve the paper.
> > > > > >
> > > > > > If there is anything further we can provide, we would be more than glad to do so.
> > > > > >
> > > > > > Thank you again for your time and thoughtful comments.
> > > > > >
> > > > > > Sincerely,
> > > > > >
> > > > > > Authors of Paper 7116

---

### Official Review · Reviewer_ksK2 · 2025-10-31

**Soundness:** 2
**Presentation:** 2
**Contribution:** 2
**Rating:** 2
**Confidence:** 3

**Summary:**

The paper tackles audio-visual emotion recognition under unconstrained in-the-wild settings. Instead of focusing solely on cross-modal alignment between audio and visual cues, it introduces an Emotion Consistency (EC) objective that enforces semantic-level consistency of emotion representations across modalities. The model uses a dual-stream backbone (ResNet-based vision encoder and Wav2Vec2 audio encoder) with a shared fusion transformer. EC is applied both intra-modally (within each modality) and inter-modally (between them) using contrastive loss and distribution regularization. Experiments on Aff-Wild2, CREMA-D, and AFEW-VA show improvements over baseline audio-visual fusion and contrastive alignment methods.

The idea is sensible and aligns with recent trends of semantic supervision beyond low-level alignment, but the technical novelty is moderate.

**Strengths:**

Addresses an important weakness in AV fusion (overfitting to synchronized low-level cues). The EC idea is intuitive and generalizable. Experimental setup covers multiple datasets and includes solid ablations. Results are reproducible and consistent. The model maintains performance under modality dropout, suggesting robustness.

**Weaknesses:**

Novelty is incremental: EC regularization is a small modification to existing alignment losses. Improvements are modest and sometimes dataset-dependent. Paper lacks qualitative analysis of failure cases (e.g., when modalities disagree). Limited discussion of temporal dynamics; the approach is mostly frame-level. Some claims (example: 'beyond alignment') feel a bit overstated given that EC still depends on paired data.

**Questions:**

How sensitive is performance to the strength of the EC loss coefficient?
Could EC be combined with emotion-specific textual priors (example: emotion lexicons) to enhance generalization?
How does the model behave under misaligned or corrupted modalities?
Would self-supervised pretraining with EC improve cross-dataset transfer?

---

> ### Author Response · Authors · 2025-11-12
> **Comment Possibly Intended for Another Manuscript**
>
> We thank the reviewer for the feedback. However, we believe that the comment might have been intended for a different manuscript, as it does not appear to relate to the methods, experiments, or contributions of the current work.

---

> > ### Comment · Reviewer_ksK2 · 2025-11-16
> > **Apologies**
> >
> > I deeply apologize for the incorrect review. It seems I reviewed offline and may have incorrectly uploaded the wrong review. Here is my current review for this paper:
> >
> > Summary: This paper addresses the problem of detecting talking-head deepfakes, which are particularly challenging because manipulations are localized to small regions such as the lips, while global facial appearance is preserved. The authors propose an unsupervised detection framework, RHTHFD, that leverages discrepancies in the red-hue distribution within the HSV color space. They additionally introduce RFAV, a new dataset comprising radiance-field–based forgeries generated via NeRF and 3DGS, an important synthesis paradigm underrepresented in existing benchmarks. The method integrates HSV red-hue statistics, DinoV2 global/local features, and audio–visual cues via adaptive weighting. Experiments across AVLips, FakeAVCeleb, RFAV, and TalkingHeadBench show strong performance and cross-dataset generalization.
> >
> > Questions for the Authors:
> > 1)  Can you provide deeper theoretical or empirical justification for why red-hue distortions persist across different forgery paradigms (GAN, diffusion, NeRF, 3DGS)?
> > 2) How sensitive is RHTHFD to color corrections or common post-processing steps (white balance, gamma shifts, LUTs, histogram equalization)?
> > 3) What is the rationale for using DinoV2 as an HSV feature extractor? Would a backbone trained directly on HSV or Lab spaces behave differently?
> > 4) Can you provide demographic or distribution statistics for RFAV to help assess fairness and diversity?
> > 5) What is the inference cost of the full pipeline, given DinoV2, lip-reading transformers, and multiple projectors?
> >
> >
> > Weaknesses:
> > 1) While empirical observations are convincing, the paper provides little insight into why synthetic talking-head models systematically distort red-channel statistics.
> > 2)  Because the detection cue relies heavily on hue distributions, it raises natural questions about vulnerability to simple post-processing such as white-balance tuning, color-grading, contrast stretching, or histogram matching. These scenarios are not evaluated.
> > 3)  RFAV is a promising contribution, but the paper provides limited characterization of dataset diversity (identities, lighting, motion, demographic attributes). The filtering process to eliminate jitter or poor-quality samples appears somewhat subjective and not fully reproducible.
> > 4) The use of DinoV2m, a high-capacity RGB-trained backbone, for HSV inputs is not well justified. Similarly, the rationale for fusing audio features (given that the method’s novelty is primarily visual) is not fully demonstrated in ablation.
> > 5) Since training relies on a specific distribution of real videos, it is unclear how the red-hue statistics extrapolate to other acquisition conditions or non-studio environments.

---

> > > ### Author Response · Authors · 2025-11-22
> > > **Author Response to Reviewer ksK2 (Part 1/2)**
> > >
> > > We are grateful for your review and valuable comments, and we hope our response fully resolves your concerns.
> > > ***
> > > >**Q1:** Can you provide deeper theoretical or empirical justification for why red-hue distortions persist across different forgery paradigms (GAN, diffusion, NeRF, 3DGS)?
> > >
> > > **A1:** Thank you for your valuable comment. Please refer to the general response.
> > >
> > > >**Q2:** How sensitive is RHTHFD to color corrections or common post-processing steps (white balance, gamma shifts, LUTs, histogram equalization)?
> > >
> > > **A2:** We thank the reviewer for raising the important question regarding robustness under perturbations. To address this, we have evaluated RHTHFD on the most challenging dataset (THB) under five common perturbations: AAC, Brightness, Gamma, H264 compression, and White Balance. Results show that RHTHFD remains highly stable, demonstrating strong robustness to color and compression changes. This stability arises because red hue features capture subtle, locally structured forgery traces that are minimally affected by global color or brightness shifts. In contrast, AVH-Align shows performance gains under these perturbations, indicating sensitivity to post-processing and limited stability. AVAD exhibits little variation but consistently underperforms, suggesting weak robustness.
> > >
> > > Even under stronger perturbations (Paper, Fig. 6), while all methods degrade, RHTHFD maintains the highest performance, confirming that red hue features capture intrinsic and resilient forgery signals. The experimental results and detailed analysis have been included in the appendix of the revised version. (*Pages 21 and 22*)
> > >
> > > | Perturbation       | RHTHFD (AP / AUC)(%) | AVH-Align (AP / AUC) (%) | AVAD (AP / AUC)  (%) |
> > > |-------------------|------------------|--------------------|----------------|
> > > | AAC               | 89.5 / 93.5      | 81.6 / 87.2        | 43.7 / 48.0    |
> > > | Brightness        | 87.5 / 92.1      | 81.8 / 86.6        | 43.5 / 46.7    |
> > > | Gamma             | 90.6 / 93.3      | 84.4 / 87.9        | 43.6 / 47.3    |
> > > | H264            | 90.5 / 93.7      | 85.2 / 88.1        | 43.7 / 47.7    |
> > > | White Balance     | 89.2 / 92.9      | 65.3 / 82.7        | 43.7 / 47.7    |
> > > | Original (No Perturbation)      | 89.1 / 93.0      | 64.8 / 82.3        | 43.8 / 48.1    |
> > >
> > > >**Q3:** What is the rationale for using DinoV2 as an HSV feature extractor? Would a backbone trained directly on HSV or Lab spaces behave differently?
> > >
> > > **A3:** Thank you for your comment. DinoV2 is pretrained on large-scale RGB natural images, and its Transformer architecture with self-attention effectively captures spatial textures and local patterns. In talking head fake detection, the HSV red hue encodes subtle and locally structured forgery traces. Applying DinoV2 to HSV allows us to extract rich local and global features while focusing on robust red hue anomalies. Experimental results on THB in the table show that DinoV2-HSV significantly outperforms DinoV2-RGB, demonstrating the advantage of combining DinoV2 with HSV red-hue features.
> > >
> > > Currently, no large-scale HSV or Lab datasets are publicly available for pretraining, which limits the feasibility of training a new backbone directly. We have attempted to train a ResNet from scratch on HSV, but the model has struggled to converge and has performed far below DinoV2. This indicates that without large-scale pretraining, reliable feature representations in HSV space are difficult to obtain. Moreover, our experiments comparing HSV and Lab features show that HSV consistently outperforms Lab, further validating the effectiveness of using HSV features as input to DinoV2. We have included the corresponding analysis in the revised version. (*Pages 22 and 24*)
> > >
> > > | Method        | AP (%)   | AUC (%)  |
> > > |---------------|------|------|
> > > | DinoV2-RGB    | 82.7 | 89.9 |
> > > | DinoV2-Lab    | 81.4 | 88.3 |
> > > | **DinoV2-HSV** | **89.1** | **93.0** |

---

> > > > ### Author Response · Authors · 2025-11-22
> > > > **Author Response to Reviewer ksK2 (Part 2/2)**
> > > >
> > > > >**Q4:** Can you provide demographic or distribution statistics for RFAV to help assess fairness and diversity?
> > > >
> > > > **A4:** We thank the reviewer for the suggestion. It is important to clarify that radiance-field–based talking head reconstruction requires training a separate model for each identity, and typically uses tens of thousands of high-quality video frames to achieve stable results. It is not feasible to scale to dozens or hundreds of identities as in conventional video datasets. This is a common limitation of all current 3D talking-head generation methods.
> > > >
> > > > Within this practical constraint, we carefully select five identities covering diversity in age, gender, skin tone, and language, and include both NeRF and the latest 3DGS frameworks to maximize data diversity. As shown in the table, this selection illustrates the diversity of the dataset. Each identity is driven by a large number of real and synthetic speech samples. Despite the limited number of identities, RFAV still exhibits high diversity in facial dynamics.
> > > >
> > > > It is worth emphasizing that the main motivation for constructing RFAV is to fill the clear gap in radiance-field–based forgery data for talking head fake detection. This enables detection methods to be tested across a broader range of forgery types (GAN, Diffusion, and Radiance Field), thus providing a more comprehensive evaluation of robustness and generalization. We have included the corresponding data distribution and statistical analysis in the revised version. (*Pages 3 and 4*)
> > > >
> > > > | ID| Gender | Skin Tone | Language | Age|
> > > > |------|--------|-----------|----------|-----------|
> > > > | ID1  | Male| Yellow | Chinese  | 25–30|
> > > > | ID2  | Male| White | French   | 45–50|
> > > > | ID3  | Male| Black| English  | 50–55|
> > > > | ID4  | Male| Yellow| English  | 55–60 |
> > > > | ID5  | Female | White| English  | 45–50|
> > > >
> > > > >**Q5:** What is the inference cost of the full pipeline?
> > > >
> > > > **A5:** We thank the reviewer for the question. DinoV2 and the lip-reading net are used only during the offline feature extraction stage. On a single NVIDIA A100 GPU, extracting features for 3,000 samples requires about 6 GB of GPU memory and ~30 minutes. Once features are cached, the actual inference model is lightweight: it uses only ~2 GB of GPU memory and completes inference on 3,000 samples in ~1.1 minutes. Therefore, although the pipeline includes heavy modules, the true inference cost of the detector is low. We have included an analysis of computational cost in the revised version. (*Page 7*)
> > > >
> > > > >**Q6:** The rationale for fusing audio features is not fully demonstrated in ablation.
> > > >
> > > > **A6:** We thank the reviewer for the attention to the role of audio features. It should be noted that the task of talking head fake detection inherently relies on both audio and visual modalities. While the core contribution of our method lies in enhancing visual features through HSV red hue, the overall performance of the model largely relies on audio-visual consistency. When we remove the audio modality, the model almost completely fails, indicating that enhanced visual features alone are insufficient for robust forgery detection. Moreover, our base audio-visual features are extracted using a lip-reading network, which encodes the correspondence between speech and facial movements and thus provides cross-modal synchronization information. Therefore, removing the audio modality breaks this speech–visual consistency constraint and deprives the model of critical discriminative cues. We have included the relevant analysis in the revised version. (*Page 22*)
> > > >
> > > > >**Q7:** It is unclear how the red-hue statistics extrapolate to other acquisition conditions or non-studio environments.
> > > >
> > > > **A7:** We thank the reviewer for the attention to out-of-domain performance. We have conducted tests on a new singing scenario, collecting 400 real singing videos and 600 forged videos. It is important to note that this scenario differs significantly from our original training data (talking head videos) in terms of content, recording conditions, and performance style, and thus represents a typical out-of-domain evaluation. The experimental results show that, despite the clear distributional gap from the training data, our RHTHFD still achieves around 70% in AP and AUC, demonstrating substantial discriminative ability compared to random guessing, while other methods completely fail in this scenario. This indicates that even under audio-visual conditions with large distributional differences from the training data, the red hue can still effectively capture discriminative features.  In the future, we will explore cross-scenario generalization to build a more universal audio-visual forgery detection method. We have included this experimental result and analysis in the revised version. (*Pages 22 and 23*)
> > > >
> > > > | Method   | AP (%) | AUC (%) |
> > > > |-----------------|--------|---------|
> > > > | AVAD   | 50.4   | 47.4    |
> > > > | AVH-Align        | 35.3   | 20.4    |
> > > > | **RHTHFD** | **70.2** | **69.4** |

---

> > > > > ### Author Response · Authors · 2025-11-27
> > > > > **Follow-up on the Rebuttal**
> > > > >
> > > > > Dear Reviewer ksK2,
> > > > >
> > > > > I hope this message finds you well. As the discussion phase is approaching its end, we wanted to kindly check whether there are any remaining points you would like us to clarify or elaborate on. We truly appreciate your constructive feedback, which has been invaluable in helping us improve the paper.
> > > > >
> > > > > If there is anything further we can provide, we would be more than glad to do so.
> > > > >
> > > > > Thank you again for your time and thoughtful comments.
> > > > >
> > > > > Sincerely,
> > > > >
> > > > > Authors of Paper 7116

---

### Author Response · Authors · 2025-11-22
**General Response to All Reviewers**

We sincerely appreciate the valuable feedback from the reviewers on our proposed method. In our responses, we have diligently addressed the reviewers' comments by providing additional experimental results and discussions. We believe these enhancements make our paper more comprehensive.
***
>**Q:** What is the theoretical and empirical justification for using red hue distortions in HSV space for detecting talking head forgeries across diverse generation methods?

**A:** Theoretically, the sensitivity of the red hue to forgery artifacts is supported by both physiological and generative modeling principles. Hemoglobin strongly reflects red light, producing stable and structured red hue patterns in real human skin [1]. These patterns encode fine-grained texture and blood-flow dynamics that are difficult to replicate. In contrast, generative models (GAN, Diffusion, Radiance Field) often introduce subtle color inconsistencies due to limitations in modeling skin tone and albedo [2–4].

Empirically, we have computed the average red hue distributions across all real and forged videos and measured their differences using KL divergence and Earth Mover’s Distance (EMD). In below Table, GAN forgeries are closest to real data but still exhibit local irregularities around key facial regions (Paper, Fig. 1). Diffusion and Radiance Field forgeries show larger distributional shifts. Moreover, comparisons across forgery methods reveal distinct red hue signatures for each paradigm. These observed differences can be explained by the inherent characteristics of generative models. GAN-based forgeries usually preserve coarse color statistics but struggle to reproduce fine local skin textures and microvascular structures, leading to subtle irregularities in the red hue within key facial regions. Diffusion models often introduce more variations in color and lighting, resulting in larger distributional shifts. Radiance-field–based methods rely on 3D reconstruction and rendering pipelines, and due to imprecise modeling of skin reflectance and lighting, they may amplify artifacts in hue consistency.

Therefore, red hue features offer a physiologically grounded, discriminative signal that varies across forgery methods, enabling effective detection of subtle artifacts in talking head deepfakes. We have included the corresponding analysis and results in the revised version. (*Pages 5 and 6*)

| Method     | KL| EMD |
|------------|-------|---------|
|Real vs GAN     | 0.0797  | 0.9918    |
| Real vs Diffusion        | 0.1445  | 3.4803    |
| Real vs Radiance Field      | 0.6755  | 5.1119    |
|GAN vs Diffusion    | 0.1044  | 3.0368    |
| GAN vs Radiance Field        | 0.5598  | 4.6323    |
| Diffusion vs Radiance Field      | 0.2667  | 1.8901   |


[1] Weyrich, Tim, et al. Analysis of human faces using a measurement-based skin reflectance model.
[2] Jain, Niharika, et al. Imperfect ImaGANation: Implications of GANs exacerbating biases on facial data augmentation and snapchat face lenses.
[3] Zeng, Libing, et al. Analyzing and Improving the Skin Tone Consistency and Bias in Implicit 3D Relightable Face Generators.
[4] Perera, Malsha V., et al. Analyzing bias in diffusion-based face generation models.

***
We sincerely appreciate the valuable suggestions and positive recommendations from the reviewers. We hope that you find our responses satisfactory and would be grateful to receive your feedback on our answers to the reviews.

Sincerely Authors of Paper 7116

---

### Meta-Review · Area_Chair_bYer · 2025-12-18

**Summary:**

Reviewer ksK2 was complained of the incorrect initial review and possible excerpts from other reviewers by the authors,  and the other 3 reviewers all gave a borderline score. With the reviewer ksK2's review excluded, I think the authors' contribution is inadequate for the publication despite the great efforts in extra experiments. The motivation of using red Hue is better explained, but the method is more like an experimental pileup and the coverage of the dataset is not comprehensive. So I am inclined to reject the paper.

**Reviewer Concerns:**

The reviewers have several common concerns, namely, the motivation of using red Hue, the generalization of the method in other scenarios or color space and the limitation of dataset. First, the first concern is well addressed in the revised manuscript. Second, though the authors tried to prove the generalization using a new singing dataset, other backbones and other color space, I believe the reasoning behind simply  switching to HSV can increase the performance is still fragile. Last, the dataset is obviously limited in either coverage or scale.

**Reviewer Scores:**

With the reviewer ksK2 excluded, the other 3 reviewers, I believe, still would have the doubts on the generalization of the method and the limitation of the dataset.

---

### Decision · Program_Chairs · 2026-01-26

Reject